# Estimating the mass of tephra accumulated on roads to best manage the impact of volcanic eruptions: the example of Mt. Etna, Italy

Luigi Mereu[1,2], Manuel Stocchi[1*], Alexander Garcia[1], Michele Prestifilippo[3], Laura Sandri[1], Costanza Bonadonna[4] and Simona Scollo[3]

[1]Istituto Nazionale di Geofisica e Vulcanologia, Sezione di Bologna, 40100 Bologna, Italy

[2]CETEMPS Center of Excellence, University of L'Aquila, 67100 L'Aquila, Italy

[3]Istituto Nazionale di Geofisica e Vulcanologia, Osservatorio Etneo, Sezione di Catania, 95015 Catania, Italy

[4]Department of Earth Sciences, University of Geneva, 1205 Geneva, Switzerland

* Now at Department of Earth and Geoenvironmental Sciences, University of Bari "Aldo Moro", 70125 Bari, Italy

*Correspondence to*: Luigi Mereu (luigi.mereu@ingv.it)

To be submitted to NHESS

**Abstract.** Explosive eruptions release significant quantities of tephra, which can spread and settle on the ground, potentially leading to various types of damage and disruption to public infrastructure, including road networks. The quantification of the tephra load is, therefore, of significant interest to evaluate and reduce environmental and socio-economic impact, and for managing crises. Tephra dispersal and deposition is a function of multiple factors, including the mass eruption rate (MER), tephra characteristics (size, shape, density), top plume height ($H_{TP}$), grain size distribution (GSD) and local wind field. In this work we quantified the tephra mass deposited on the main road network on the east-southeast flanks of Mt. Etna (Italy) during lava fountains occurred in 2021, which reached heights of hundreds of metres. We focused on road connections of municipalities significantly affected by these events such as Milo, Santa Venerina and Zafferana Etnea. First, we analysed a sequence of 39 short-lasting and intense Etna's lava fountains detected by the X-band weather radar, applying a volcanic ash radar retrieval approach that permits us to compute the main eruption source parameters (ESPs), such MER, $H_{TP}$ and GSD. When radar measurements were unavailable for a specific event, we analysed images acquired both by the SEVIRI radiometer and by the visible and/or thermal infrared camera of the Istituto Nazionale di Geofisica e Vulcanologia, Osservatorio Etneo (Catania) to derive the main ESPs. Second, we used the computed ESPs as inputs to run two different numerical models, Tephra2 and Fall3D, and reproduce tephra dispersal and accumulation on the road network. Finally, we produce, for the first time, georeferenced estimates of tephra mass deposited on the whole road network of three municipalities, allowing us to identify the main roads which have been mostly impacted by tephra accumulation, as well as

to estimate the total mass of primary tephra that has been removed from roads. Such information represents a valuable input
for planning and quick management of the short-term tephra load hazard for future Etna explosive events.

## 1 Introduction

The estimation of tephra mass deposited on the ground, following a volcanic explosive eruption, remains a key information
that is not well-covered in the literature. In fact, tephra dispersal and fallout is by far the most widespread volcanic hazard
affecting both local and distal areas (Jenkins et al. 2015; Barsotti et al., 2018; Bonadonna et al. 2021b) including impact on
public health (Baxter, 1990; Horwell and Baxter, 2006), roofs/building collapse (Spence et al., 2005), dangerous road
conditions (Blong, 1996; Wilson et al., 2012; Jenkins et al., 2014; Blake et al., 2017), contamination of water reservoirs and
vegetation (Wilson et al., 2012; Ágústsdóttir, 2015), damages to electrical infrastructure (Bebbington et al., 2008; Wardman
et al., 2012; Wilson et al., 2012, Dominguez et al., 2021), transportation system disruptions (Casadevall, 1994; Guffanti et
al., 2009; Wilson et al., 2012), and impact on telecommunication networks (Wilson et al., 2012). Even tephra associated
with relatively small intensity eruptions may induce various disrupting effects on transport infrastructure such as aeroplane
engine failure and visibility reduction during both primary tephra fall and ash remobilisation (Sarna-Wojcicki et al., 1981;
Bonadonna et al., 2021b; Johnston and Daly, 1995; Wilson et al., 2014). In particular, tephra accumulation, although not
causing significant physical damage on the road network, can cause wide disruption including reduction of skid resistance,
obscuration of road markings and damage to car air filters (Blake et al., 2016, 2017). Tephra particles are also very abrasive
with the degree of abrasiveness dependent on the hardness of the material forming the particles and their shape and
angularity (Blong, 1984; Johnston, 1997; Labadie, 1994; Heiken et al., 1995; Miller and Casavedall, 1999; Gordon et al.,
2005; Wilson et al., 2012; Blake et al., 2017).
During volcanic eruptions, routes may be required for the evacuation of residents and to allow emergency services and civil
protection personnel to access the affected areas. Road networks are critical for society under normal operating conditions
and especially during emergencies (e.g. Bonadonna et al. 2021a; Hayes et al. 2022), for both immediate and long-term
recovery, including clean-up and disposal of pyroclastic material, and restoration of services and commerce (Blake et al.,

52  2017).

In this work, for the first time, we quantify the tephra mass accumulated on the road network of east and south-east sectors
of Etna, which were preferentially affected during the sequence of explosive events of 2021. Usually, the eruptive sequences
at Etna are characterised by short-lasting explosive events, with duration of few hours, separated by periods that can last
from few hours to few days (Calvari et al., 2018; Andronico et al., 2021; Calvari et al., 2022a). Most studies on exposed
critical infrastructure have focussed on larger events and tephra-fallout accumulations greater than 10 kg/m$^2$ (Wilson et al.,
2012, Blake et al., 2017; Scollo et al., 2013). However, areas around Etna are more frequently impacted by rates and
volumes of smaller tephra accumulation (Scollo et al., 2013). Limited quantitative data available for explosive activity have
hampered a reliable quantification of the impact of the tephra deposition at Etna. To investigate its impact on road networks
and better characterise its behaviour, we analysed a sequence of several lava fountains occurred between February 2021 and
October 2021, focusing our analysis on 39 events that generated volcanic plumes dispersed by wind mostly towards the
east-southeast flanks of the volcano (direction between 90 degrees and 130 degrees from North). These episodes began at
the South East Crater (SEC) as initial Strombolian activity that, with time, evolved to lava fountain activity, also named
paroxysm.
Remote sensing is routinely used for monitoring the eruptive activity of Etna; the Istituto Nazionale di Geofisica e
Vulcanologia, Osservatorio Etneo (INGV-OE) runs a remote sensing network that includes both ground-based (such as
thermal infrared and visible cameras) and satellite-based sensors (Scollo et al., 2019). An X-band weather radar located in
Fontanarossa airport (Catania), which is part of the monitoring network of the Italian Department of Civil Protection (DPC)
allows to monitor and analyse the Etna's eruptions as well (Marzano et al. 2020; Mereu et al. 2020). Using these sensors, we
can observe in almost all the cases the temporal evolution of explosive activity and characterise it quantitatively in terms of
mass eruption rate (MER) and top plume height ($H_{TP}$). These two parameters are among the main input variables for
advection-dispersion models (Scollo et al., 2008a; Biass et al., 2017; Tadini et al., 2022; Takishita et al., 2021), such as
Tephra2 and Fall3D (Bonadonna et al., 2005; Costa et al., 2006; Folch et al., 2009, 2020), which have been used in this
work to simulate the tephra dispersion and calculate the deposit load at the ground. For each of the 39 events, we estimate
the tephra deposited on the road network in order to identify the roads mostly exposed to tephra accumulation and both to
evaluate the ground mass load that is expected to be removed and the costs involved in clean up operations. The deposited
tephra creates disruptions especially on main roads, considering the large stretches of roadway that may face hazardous
driving conditions. Moreover, analysing the simulations done using both models, we investigate their sensitivity to
variations of tephra granulometric characteristics and assess the associated uncertainties. The numerical output from a single
simulation is a georeferenced map of tephra load, useful to analyse the impact of deposited tephra fallout on roads (Scollo et
al., 2009; Scollo et al., 2013; Costa et al., 2012; Bonadonna et al., 2005; Barsotti et al., 2018; Bonadonna et al. 2021a).
Studying the lava fountain events of Etna in 2021 provides valuable insights for future planning during similar events. By
examining the past eruptions, we can predict tephra deposits, providing useful information to decision makers to develop
better cleaning up strategies. Indeed, for future eruptions, it is important to consider creating a model that uses real-time data
to improve predictions and clean up plans. Overall, this research can help crisis management and enhance safety in volcanic
regions. Additionally, recent regional legislation (DA n. 8/Gab. 22/04/2024, https://www.regione.sicilia.it) permits the use
of the volcanic tephra for building applications; in this new framework, this study represents an initial effort to estimate the
volume of tephra that may be reused rather than disposed of, transforming a potential problem into a resource.
The work is organised as follow: Section 2 provides a brief description of Etna's lava fountain activity; Section 3 presents
ground- and satellite-based sensor data, along with the main eruption source parameters (ESPs) retrieved from them, and the
methodology employed to analyse the model results; Section 4 validates the results against previously published data for a
specific event (February 28th, 2021); Section 5 discusses the modelling outputs and provides conclusive remarks.
**2 The lava fountains at Etna in 2021**
Etna, one of the most active volcanoes in the world, is a stratovolcano on the east coast of Sicily, Italy. It rises to more than
3,300 m in altitude and has four main summit craters: North-East Crater (NEC), Voragine (VOR), Bocca Nuova (BN) and
South-East Crater (SEC). Between February 2021 and October 2021, approximately sixty paroxysmal episodes occurred
from the SEC. Of these, 39 episodes lasted several hours and were dispersed towards the E-SE (Calvari et al., 2021). Lava
fountains, which are formed by a hot inner core consisting of a mix of liquid clots, pyroclasts, and magmatic gases, are often
observed during paroxysmal episodes at Etna and can rise several hundred metres above the volcanic vent (Wilson et al.,
1995; Taddeucci et al., 2015). Moreover, during those events it is likely that an eruption column of almost 10-15 km above
sea level can form above the lava fountains. Usually, lava fountains are divided in three phases (e.g., Alparone et al., 2003;
Mereu et al., 2020; Calvari et al., 2022b): 1) Resumption phase, which commonly begins with slow initial effusion from the
vent, followed by a progressive increase in explosive activity; 2) Paroxysmal phase, lasting from 10 to 120 minutes, during
which there is a rapid transition from Strombolian activity to sustained lava fountains that rise up an altitude of 2-6 km
above the vent; 3) Conclusive phase, during which the eruptive episode gradually ceases.
**3 Methods**
**3.1.1 Sensors and outputs**
For the analyses in this work we select all the Etna eruptions characterised by a tephra plume dispersed towards east-
southeast flanks of the volcano. These events, listed in Table 1, have been observed using different sensors that are briefly
described below:
a) X-band Weather Radar (XWR), located in the airport of Catania, 32 km at SSE Etna summit craters (Figure 1). The
scanning agility in elevation and azimuth of this sensor allows it to probe the tephra cloud in any weather condition and both
during the day and night (Mereu et al., 2022, 2023; Montopoli, 2016; Vulpiani et al., 2016). Applying the Volcanic Ash
Radar Retrieval (VARR) methodology (e.g. Marzano et al., 2012, 2020; Mereu et al., 2015, 2020) to measure radar
reflectivity factor, we estimate: i) the top plume height $H_{TP}$ (km) above sea level, which is the maximum height reached by
the eruption column, calculated as the maximum altitude of the radar-detected volume above the volcanic vent contaminated
by the minimum detectable tephra concentration; ii) the mass eruption rate $Q_M$ (kg/s), that is estimated by the time-space
variation of tephra concentration detected above the Etna summit when probing the volcanic plume; iii) the total erupted
mass TEM (kg), which is the total mass of pyroclastic material erupted during the explosive event; iv) the ash-fall rate Ra
$(kg/m^2 \cdot h)$ which is useful to derive the tephra load integrating this quantity over a time interval and assuming that the radar
measurements closer to the ground are indicative of tephra deposited on the ground from the vertical column above a
considered position (Mereu et al., 2015).
b) Etna Catania Visible calibrated camera (ECV), located in Catania about 30 km from Etna summit craters (Scollo et al.,
2019; Aravena et al., 2023, Figure 1); it allows us to monitor the altitude of dispersed plume during the light hours when the
visibility is not compromised by the meteorological cloud cover. In this way, we can derive the time sequence of $H_{TP}$.
c) Spinning Enhanced Visible and InfraRed Imager (SEVIRI), on board of Meteosat Second Generation (MSG) satellites, is
a multispectral radiometer which produces daytime brightness temperature (BT) images with 3 km resolution. Selecting the
BT along the Etna summit in the channel of 10.8 μm, that is more sensitive to the tephra dielectric signature, we infer $H_{TP}$
looking for the altitude in which the detected BT can be found in the temperature profile as a function of altitude, which is
derived from the hydro-meteorological service of Agenzia Regionale Prevenzione e Ambiente (ARPA) in Emilia Romagna
(Scollo et al., 2009; Romeo et al., 2023).
d) Etna Nicolosi Thermal (ENT) infrared camera, located in Nicolosi at about 15 km from Etna summit (Figure 1), which
measures the thermal activity associated with lava fountains. It is worth highlighting that when the radar measurements were
not available and the volcanic plume was not easily detectable by the satellite sensor or by the visible calibrated camera,
analysing the ENT images we have identified the Incandescent Jet Region (IJR), which is a proxy of the lava fountain
height. As described in Mereu et al., (2020), the time sequence of maximum height of IJR area can be converted in exit
velocity $v_{ex}$(m/s) of pyroclastic material, using the Bernoulli equation under the following approximations: i) most of the
pyroclastic material is sufficiently large to be considered as accelerated projectiles confined in this IJR; ii) atmospheric
density variations and drag effects are negligible. Assuming a trustworthy value of tephra-gas mixture density and of surface
vent, we can deduce $Q_M$ the surface flow approach (SFA) described in Marzano et al., (2020) and Mereu et al., (2022). It is
worth noting that $H_{TP}$ obtained from various sensors, such as XWR, ECV, and SEVIRI, exhibits a comparable time trend, as
shown in Freret-Lorgeril et al. (2021) and Scollo et al. (2019). The complete ESPs dataset for each of the Etna events
considered in this study is displayed in Table 1. Real-time estimation of ESPs can be challenging, especially during the
initial phase, increasing the uncertainty in short-term forecasts of plume dispersal (Scollo et al., 2008a). Furthermore, we
also collected qualitative information about the eruptions, such as plume height (based on VONA reports; Scollo et al.,
2019; Corradini et al., 2018), the presence of tephra fallout and the start and end times of Strombolian and lava fountain
activities (based on bulletins and reports available on the INGV-OE website (www.ct.ingv.it).

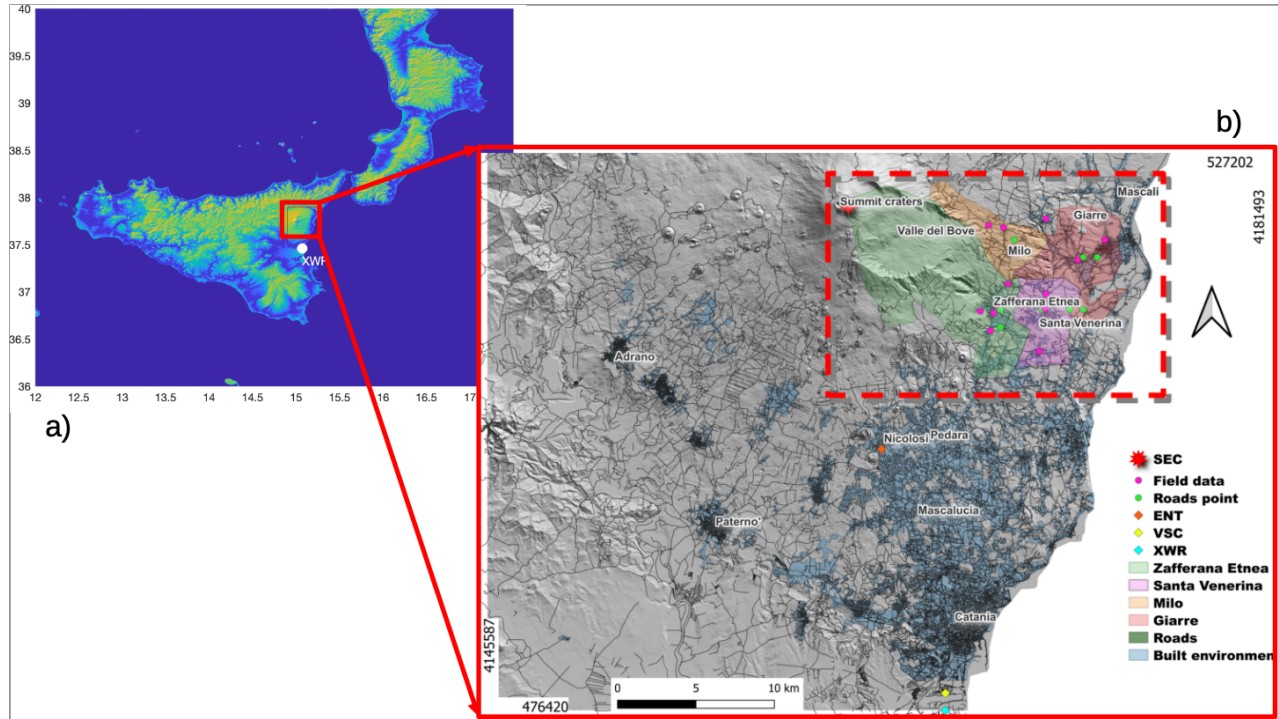


**Figure 1: (a)** Map of South Italy (Sicily and Calabria), with the red square showing the Etna area framed in the right panel; **(b)** georeferenced map of road network (dark lines) of the Etna volcano area(shapefiles with the road data is publicly available from the Regione Sicilia website: https://www.regione.sicilia.it/). The UTM coordinates (area 33S) are shown in the lower-left part and in the upper-right part of the picture, respectively. The areas of four municipalities, of which three are under examination, and the built environment are highlighted with different colours, whereas each sensor, field data and road points are identified by coloured symbols as listed in the legend in the right side: ground-based sensors employed in this work (the visible camera VSC, the thermal infrared camera ENT and the X-band radar XWR); the South East Carter SEC; 14 field data as derived by Pardini et al., (2023); 8 road points. The rectangle highlighted with the dotted red line identifies the area examined in detail in Figures 3 and 7.

On the other hand, each sensor previously described allows us to measure some features of the lava fountains, which need further elaborations to obtain the ESPs. In this work, $H_{TP}$ (km) and $Q_M$ (kg/s) quantities are directly derived by processing XWR measurements. When it was not possible to determine $H_{TP}$ from XWR, ECV frames or SEVIRI data, we used the ENT images to retrieve the $Q_M$ estimates applying the SFA (Marzano et al., 2020; Mereu et al., 2022). Integrating the SFA in time, we obtain TEM (kg), whereas applying the empirical relation of Mastin et al. (2009), which links $Q_M$ to $H_{TP}$, we get the $H_{TP}$ above the Etna summit crater (which is located about 3357 m above sea level). We used the inverse Mastin equation in those cases where the $H_{TP}$ was derived from VSC or SEVIRI imagery to obtain the $Q_M$ time sequence. The starting and ending time for each Etna explosive eruption can be straightly inferred from: i) XWR-based $Q_M$ estimates, with $Q_M > 5 \ 10^5$ kg/s; ii) temporal range where ENT camera identifies the lava fountaining feeding the explosive phase; iii) time range in which a quick development of the volcanic cloud is observed by the ECV frames or SEVIRI images. Regarding the wind data, while for Tephra2 simulations we consider a horizontally-constant wind field computed as the mean wind velocity between the Etna summit craters and the maximum value of the $H_{TP}$ sequence, for Fall3D simulations we use the whole meteorological profile. This data, which is used to feed the ash dispersion models, are derived from the European Centre for Medium-Range Weather Forecasts (ECMWF) ERA5 reanalysis (https://www.ecmwf.int/en/forecasts/datasets/reanalysis-datasets/era5). The grain-size distribution usually refers to the volcanic particle size, indicated by the relation $\phi = -\log_2 (D)$, where D stands for sphere-equivalent mean diameter (measured in mm). The parameter $\phi$ refers to the whole deposit, assumed as a Gaussian distribution characterised by a maximum, minimum, mean and standard deviation equals to -6, 10 and 3, respectively. In order to consider all possible cases, in this work we vary the median $\phi$ value between -1 to +1 with a step of 0.5. The georeferenced location and the elevation of the SEC are considered to complete the set of input parameters used, as listed in Table 1. It is worth noting that for each event listed in Table 1, we consider the UTM coordinates of the vent in easting (500024.03 m) and northing (4177699.5 m), or, equivalently, in longitude (15.000273°E) and latitude (37.746592°N). We repeat each simulation, varying the median $\phi$ values, so that we obtain a total of 195 simulations from each numerical model.

**Table 1. Input parameters used for setting the numerical dispersion model Tephra2 and Fall3D: starting time and ending time of paroxysm (dd:mm:yy, hh:mm), duration $\Delta t$ (s), top plume height $H_{TP}$ (m) above sea level (a.s.l.) and above volcano vent (a.v.v.), total erupted mass TEM (kg).**

| START TIME Date T0 UTC | END TIME Date T=T0+Dt UTC | Δt [s] | H$_{TP}$ (a.s.l) [m] | H$_{TP}$ (a.v.v.) [m] | TEM [10$^7$ kg] |
|---|---|---|---|---|---|
| | | | | | |
| 17/02/21 23:40 | 18/02/21 01:20 | 6000 | 9300 | 5943 | 18 |
| 19/02/21 08:40 | 19/02/21 10:30 | 6600 | 10000 | 6643 | 29 |
| 28/02/21 07:50 | 28/02/21 09:50 | 7200 | 11900 | 8543 | 250 |
| 07/03/21 06:20 | 07/03/21 07:50 | 5400 | 11600 | 8243 | 110 |
| 12/03/21 05:50 | 12/03/21 10:50 | 18000 | 10500 | 7143 | 175 |
| 14/03/21 23:20 | 15/03/21 02:20 | 10800 | 10957 | 7600 | 540 |
| 17/03/21 02:50 | 17/03/21 05:10 | 8400 | 6300 | 2943 | 22 |
| 19/03/21 08:40 | 19/03/21 10:20 | 6000 | 10400 | 7043 | 98 |
| 19/05/21 03:00 | 19/05/21 04:30 | 5400 | 5000 | 1643 | 0.09 |
| 22/05/21 20:20 | 22/05/21 22:40 | 8400 | 11057 | 7700 | 445 |
| 24/05/21 20:30 | 24/05/21 22:45 | 8700 | 11000 | 7643 | 468 |
| 28/05/21 06:20 | 28/05/21 07:50 | 5400 | 10857 | 7500 | 0.4 |
| 28/05/21 18:10 | 28/05/21 21:10 | 10800 | 10757 | 7400 | 486 |
| 30/05/21 03:00 | 30/05/21 06:00 | 10800 | 7500 | 4143 | 46.9 |
| 02/06/21 08:10 | 02/06/21 10:50 | 9600 | 7600 | 4243 | 13.2 |
| 04/06/21 16:40 | 04/06/21 18:40 | 7200 | 7500 | 4143 | 10 |
| 12/06/21 18:30 | 12/06/21 19:10 | 2400 | 9000 | 5643 | 17.7 |
| 14/06/21 21:40 | 14/06/21 22:30 | 3000 | 6300 | 2943 | 56 |
| 16/06/21 10:30 | 16/06/21 13:00 | 9000 | 8000 | 4643 | 15.8 |
| 17/06/21 22:40 | 17/06/21 23:55 | 4500 | 12457 | 9100 | 290 |
| 20/06/21 22:40 | 21/06/21 00:40 | 7200 | 10900 | 7543 | 18 |
| 22/06/21 03:30 | 22/06/21 04:20 | 3000 | 8000 | 4643 | 11.8 |
| 23/06/21 02:00 | 23/06/21 03:40 | 6000 | 7300 | 3943 | 77 |
| 23/06/21 17:40 | 23/06/21 19:00 | 4800 | 11500 | 8143 | 120 |
| 24/06/21 09:20 | 24/06/21 11:00 | 6000 | 12200 | 8843 | 4.2 |
| 25/06/21 18:20 | 25/06/21 19:40 | 4800 | 10664 | 7307 | 4.8 |
| 25/06/21 00:30 | 25/06/21 02:40 | 7800 | 10616 | 7259 | 230 |
| 26/06/21 15:20 | 26/06/21 17:20 | 7200 | 9000 | 5643 | 22 |
| 27/06/21 08:50 | 27/06/21 10:00 | 4200 | 10000 | 6643 | 72.9 |
| 28/06/21 14:30 | 28/06/21 15:40 | 4200 | 10000 | 6643 | 68.8 |
| 01/07/21 23:40 | 02/07/21 01:40 | 7200 | 11109 | 7752 | 396 |
| 04/07/21 15:00 | 04/07/21 17:50 | 10200 | 8200 | 4843 | 8.8 |
| 06/07/21 22:00 | 06/07/21 23:45 | 6300 | 10000 | 6643 | 190 |
| 20/07/21 06:20 | 20/07/21 08:30 | 7800 | 11800 | 8443 | 79 |

| | | | | | |
|---|---|---|---|---|---|
| 31/07/21 21:00 | 31/07/21 23:50 | 10200 | 11000 | 7643 | 309 |
| 09/08/21 02:00 | 09/08/21 04:40 | 9600 | 12000 | 8643 | 140 |
| 29/08/21 16:40 | 29/08/21 18:00 | 4800 | 9000 | 5643 | 13.1 |
| 21/09/21 07:30 | 21/09/21 09:20 | 6600 | 10900 | 7543 | 47 |
| 23/10/21 08:40 | 23/10/21 11:30 | 10200 | 12300 | 8943 | 240 |

## 3.2    Models

### 3.2.1    Modelling tephra fallout

In this study, we simulate the transport, dispersal and deposition of tephra with two different numerical Eulerian models: the semi-analytical model Tephra2 and the full computational model Fall3D. We run both models on a grid covering the area [37-38.5° N, 14.5-16° E] with a spatial resolution of ~500 m. Tephra2 allows to evaluate the ground tephra deposition employing the advection-diffusion theory (Bonadonna et al., 2005, 2006; Connor and Connor., 2006; Volentik et al., 2009; Biass et al., 2016, 2017) taking as inputs: $H_{TP}$; TEM; $\phi$; the density of lithics and juveniles (volcanic particles released from the column, which varies widely from ~500 kg/m$^3$ in highly vesicular clasts to ~2700 kg/m$^3$ in dense ones); the diffusion coefficient (K), which accounts for atmospheric processes including atmospheric diffusion and cloud gravitational spreading; the fall time threshold (FTT), an empirical threshold that defines the transition between two different laws of atmospheric diffusion and the plume ratio (PR), a factor describing the mass distribution in the plume, a horizontally uniform wind field. In the Tephra2 model, it is assumed that a vertical eruption column forms above the vent. The column is discretized, and particles fall from every part of its height. The total tephra mass is vertically distributed within the eruption column according to a probability density function that represents the mass as a function of height. The model provides three different mass distributions: uniform, log-normal, and beta distribution. The total grain size distribution for the eruption is estimated assuming a normal distribution in $\phi$ units (Bonadonna et al., 2005).

Fall3D (v8.0, Folch et al., 2020) models both the particle concentration in the atmosphere (i.e. tephra cloud evolution) and the particle loading at ground level, based on a 3-D time-dependent Eulerian scheme (Costa et al., 2006; Folch et al., 2009, 2012). The model solves the advection-diffusion-sedimentation (ADS) equation over a topographical 3D domain, with several modelling options, including particle aggregation and source terms, among others. The meteorological data used in the simulations are obtained by interpolating the outputs of a meteorological model into the simulation domain. The simulations conducted for this work were performed with no aggregation, using the one-dimensional buoyant plume theory (Folch et al., 2016) as a model for the source term. The meteorological data were retrieved from the ECMWF ERA5-Reanalysis database (Hersbach et al., 2018). This methodology has the potential to track the evolution of particle concentration during an eruption, but the main limitation is the computational cost (Costa et al., 2006). Fall3D uses the same ESP's inputs listed in Table 1, but instead of TEM, it considers $Q_M$. Figure 2 shows the simulated tephra load (kg/m$^2$) maps for two among the largest-TEM eruptions ($10^8$ kg and $10^9$ kg), assuming $\phi$=0.5. Generally, we note a greater spreading of tephra deposition to the ground in Tephra2 simulations with respect to Fall3D. Both numerical models generate output files in netCDF format (e.g. https://www.unidata.ucar.edu/software/netcdf/) containing geo-referenced data on tephra load

217 (kg/m²) on the ground in UTM coordinates (zone 33-S for Etna) with a grid spacing of 500 m. The limitations of both
218 models, as a function of variation of input parameters listed in Table 1, as well as and considerations of topography, column
219 mass distribution models, bulk particle shape, and particle terminal fall velocity, are described in detail in Scollo et
220 al.(2008b).

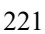

**Figure 2: Maps of tephra load (kg/m²) for the Etna lava fountain event of 17 February 2021 (a)-b)) and of 23 October 2021 (c)-d)), using the Tephra2 (a)-c)) and Fall3D (b)-d)) models. Tephra deposition is plotted as uniform iso-mass contour lines in**

**grayscale, ranging from black for values lower than 10 kg/m² to light white for values greater than 5·10³ kg/m². The colour**
**scale is the same in all panels. Global Self-Consistent Hierarchical High-Resolution Geography (GSHHG) map of Matlab.**

### 3.2.2    Calculating the tephra load and mass on the road network

Since our main interest in this work is to calculate the tephra load in the road network, we increase the grid spacing of
the tephra load data to 5 m using linear interpolation. Afterwards, we use the Quantum Geographical Information System
(QGIS) tool to determine the intersection between the downscaled tephra load data and the areas covered by the road
network. While the tracks of road network in the study area is publicly available (geospatial vector data in shapefile format,
as shown in Figure 1, from the Regione Sicilia; website https://www.regione.sicilia.it/), an accurate estimate of area of roads
is not still available. In order to estimate the area of the roads, we selected several roads in the study area and measured their
width using both QGIS integrating an OpenStreetMap layer (https://wiki.openstreetmap.org/wiki/QGIS), and satellite
images from GoogleEarth (https://earth.google.com/web/), obtaining an average width of approximately 6 m. Assuming that
the roads are generally composed of two carriageways, this value is in agreement with the prescribed widths for urban and
extra-urban roads according to the Italian law on roads (art. 2 del Testo Unico, which ranges from 2.8 m and 4 m per
carriageway). To consider an uncertainty in this gross measurement, we set the road width to $6 \pm 0.5$ m. The case study is
focused on the road networks located within the municipal area of Milo, Santa Venerina and Zafferana Etnea towns. For a
given road segment we calculate the corresponding road area (m²) and then using the tephra load (kg/m²) we calculate the
total mass (kg) deposited on each road segment.  Figure 3 (which covers the area in the rectangle shown with a dotted red
line in Figure 1) shows the simulated tephra load (kg/m²) on the geo-referenced road network within the Milo (light orange
area), Santa Venerina (light pink area) and Zafferana Etnea (light green area) municipalities for the event on 28 February
263  2021.


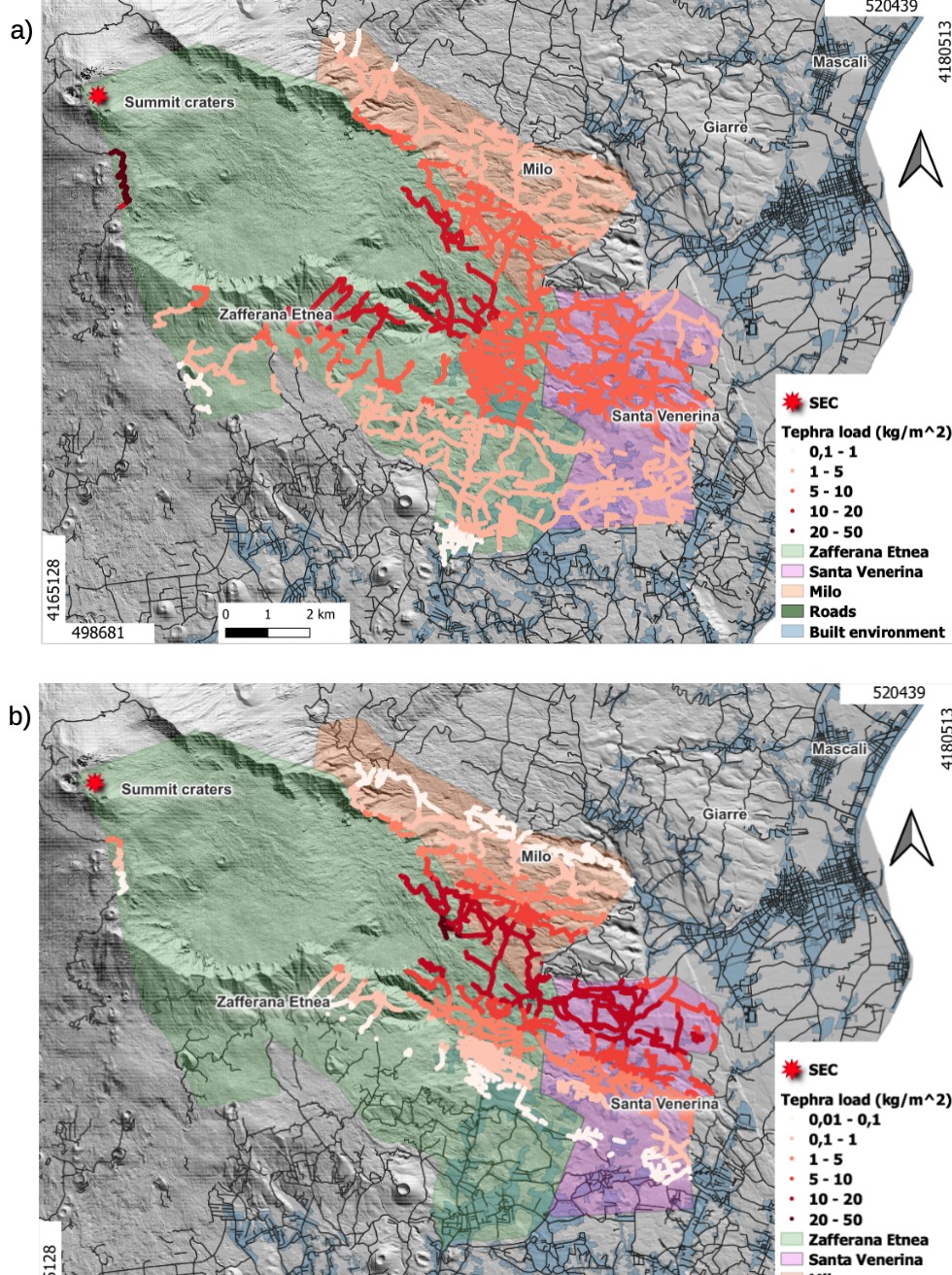


**Figure 3: Tephra load (kg/m²) on the road network of Milo, Santa Venerina and Zafferana Etnea municipalities computed for the Etna explosive event on 28 February 2021, assuming φ =0.5 and using both Tephra2 (a) and Fall3D (b) models. The tephra deposits in the road graph are shown in red scale for selected threshold levels of the tephra load values (as shown in the legend). Georeferenced map of road network (dark lines) of the Etna volcano area (shape-files with the road data is publicly available from the Regione Sicilia website: https://www.regione.sicilia.it/).**

## 4. Results

### 4.1 Validation: case study on February 28, 2021

To verify the results obtained in this study, we focus our attention on the case study of the event on 28 February 2021, which has been observed by the XWR and analysed by Pardini et al., (2023). Table 2 shows the coordinates (longitude, latitude) of 14 points in which field measurements of tephra load are available (from literature, e.g., Pardini et al., 2023) in the selected municipalities (see Figure 1) as well as the results of tephra load derived from the Tephra2 and Fall3D models, fixing the parameter $\phi$ to 0.5. The XWR retrievals are obtained considering the tephra load rate (kg/m$^2$ s) related to the first four elevations, which is equal to a few km of altitude with respect to the ground, and integrating it for the whole time sampling of the radar. In this way we can retrieve the tephra load (kg/m$^2$) related to the whole lava fountain. The correlation between the ground field data (Pardini et al., 2023) and the estimated values using Tephra2, Fall3D, and XWR are plotted, respectively, in Figure 4a, 4b, and 4c. We observe that taking as reference the observed values reported by Pardini et al., (2023), Tephra2 tends to overestimate while Fall3D tends to underestimate the tephra load values. This is evident from most of the points dispersed above and below the bisector, as well as from the differences in the slope of the regressive curves with respect to the bisector shown in Figures 4a and 4b. These discrepancies could be due to different dispersal settings used in the numerical model. In contrast, a good correlation between field data and XWR data is observed (Figure 4c), where the points mostly distribute around the bisector and the regressive straight line is almost parallel to it. To evaluate the degree of agreement between field data and tephra load estimates, we implemented a non-parametric test (namely the Kendall's tau correlation). Table 3 summarises Kendall's tau coefficients, the p-values for testing the null hypothesis of no correlation against the alternative hypothesis of a non-zero correlation; moreover, we estimate the mean absolute percentage error (MAPE) between predicted/estimated data and the observed field data. A Kendall's tau value closer to 1 indicates a better correlation between the field data and the model/estimate data, whereas a low p-value (<1%) indicates a significant correlation. This test confirms that the estimates of both XWR and Tephra2 exhibit a better correlation with the field measurements, as indicated by the respective higher tau and lower p-values. The MAPE calculation further supports this conclusion.

**Table 2. Etna eruption on February 28, 2021: tephra load (kg/m$^2$) on 14 sites, each one identified by the latitude and longitude, as deduced by Pardini et al., (2023), data computed from Tephra2 and Fall3D models (fixing $\phi$ =0.5) and retrieved by XWR are also included.**

| Coordinates (degrees) | | Tephra load (kg/m²) | | | |
|---|---|---|---|---|---|
| Longitude | Latitude | Tephra2 | Fall3D | XWR | Field data |
| 15.102649 | 37.677930 | 5.0 | 0.0 | 2.4 | 1.0 |
| 15.095485 | 37.689185 | 9.0 | 0.0 | 4.1 | 3.3 |
| 15.104990 | 37.688067 | 8.9 | 0.2 | 4.3 | 3.3 |
| 15.107469 | 37.692398 | 8.7 | 1.1 | 7.6 | 6.2 |
| 15.115397 | 37.704832 | 9.1 | 16.8 | 8.4 | 4.7 |
| 15.117104 | 37.722241 | 4.0 | 1.9 | 6.7 | 4.3 |
| 15.112271 | 37.737129 | 1.6 | 0.0 | 3.5 | 2.4 |
| 15.101504 | 37.738418 | 1.6 | 0.1 | 3.5 | 2.1 |
| 15.143073 | 37.742075 | 0.2 | 1.7 | 1.6 | 0.1 |
| 15.165928 | 37.718516 | 1.3 | 0.0 | 2.3 | 1.5 |
| 15.141991 | 37.690479 | 6.4 | 10.5 | 3.4 | 3.9 |
| 15.138243 | 37.666113 | 4.4 | 0.0 | 1.0 | 1.0 |
| 15.142994 | 37.698936 | 5.6 | 13.1 | 3.1 | 4.6 |
| 15.185512 | 37.729891 | 0.2 | 1.4 | 0.5 | 0.5 |

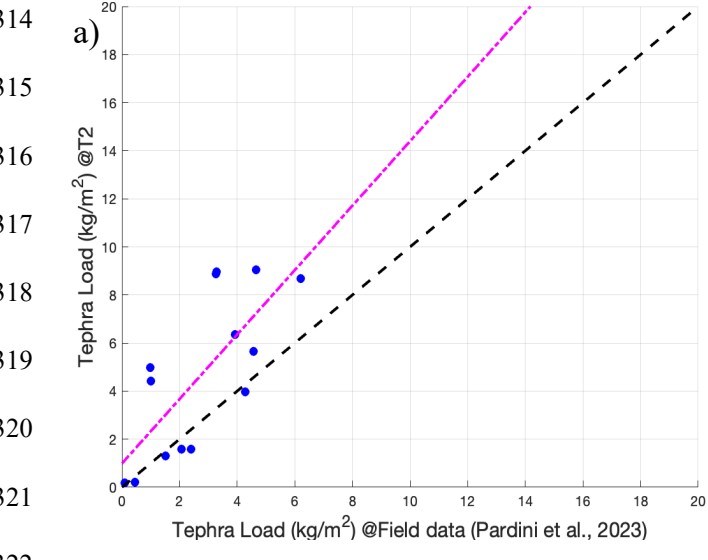
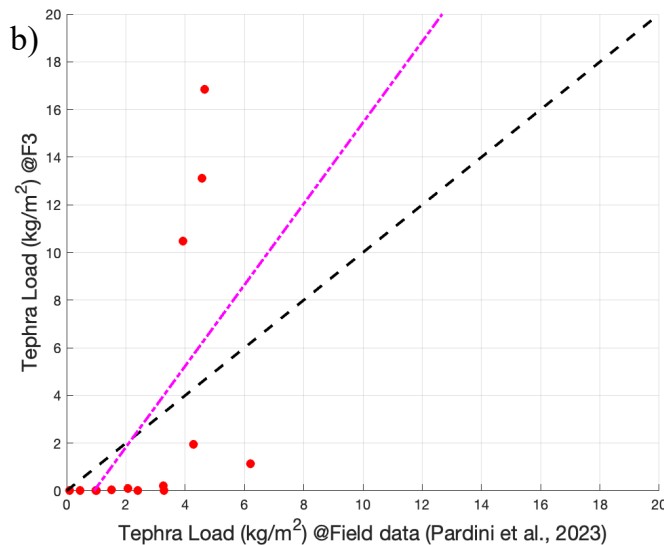










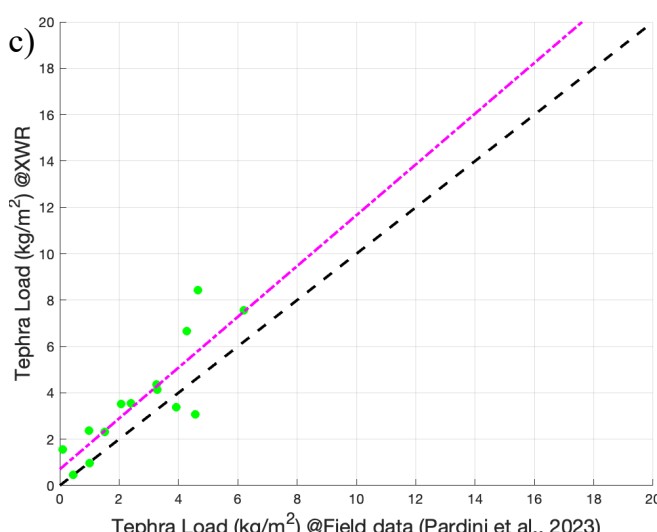

**Figure 4: Tephra load correlation: the coloured points represent the field data (Pardini et al., 2023) as a function of the results**
**from Tephra2, Fall 3D and XWR results (blue, red and green dots), as listed in Table 2 and respectively shown in panel a), b), and**
**c). The dashed dark represents the bisector, while the magenta line represents the regressive straight line.**

**Table 3. Kendall's tau correlation coefficient, p-value, and Mean Absolute Percentage Error (MAPE) computed for the Etna**
**eruption on February 28, 2021.**

| Kendall method | $\tau$ | p-value | MAPE |
|---|---|---|---|
| Tephra2-Field data | 0.73 | $2.8 \cdot 10^{-3}$ | 48 |
| Fall3D-Field data | 0.54 | $4.56 \cdot 10^{-2}$ | $3.7 \cdot 10^4$ |
| XWR-Field data | 0.84 | $1.3 \cdot 10^{-4}$ | 34 |


Assuming the width of $(6 \pm 0.5)$ m for each road, we convert the tephra load to tephra mass (kg) for the event of 28
February 2021, assuming $\phi = 0.5$. We selected a few roads, characterised by their larger extension, for each municipality:
Via Bellini and Corso Italia in Milo; via Mazzini, via Galimberti and via Stabilimenti in Santa Venerina; via Libertà, via
Zafferana Milo and via delle Rose in Zafferana Etnea. We summarise in Table 4 the total tephra mass for streets of Milo,
Santa Venerina and Zafferana Etnea as derived from Tephra2 and Fall3D models. The total mass computed for the selected
roads in Milo by Tephra2 and Fall3D shows a comparable order of magnitude ($10^7$ kg), whereas for two selected roads in
Zafferana Etnea reveals a disagreement of one order of magnitude. However, the values computed for all municipalities
show a maximum variability between approximately five times the tephra deposit collected on the ground (Scollo et al.,
2007). The Tephra2 outcomes are generally larger than those of Fall3D, with the ratio between Fall3D and Tephra2 ranging
from 0.3 to 1.4 for the roads and from 0.3 to 1.1 for towns. The variability in road width gives rise to an uncertainty in
tephra mass of approximately $\pm 9$ %. The last three rows in Table 4 show the total mass computed on the whole road
network of the three municipalities, highlighting the comparable results (same order of magnitude of about $10^9$ kg). It is

worth noting that the total mass derived from Tephra2 in Zafferana Etnea municipality is four times larger than the total mass derived from Fall3D. This variability can be observed in Figure 3. Not all roads are affected by tephra load, as shown in Figure 3b, and this can be attributed to dissimilarity in the dispersal laws implemented in each model (Scollo et al., 2008b; Bonadonna et al., 2005; Folch et al., 2012, 2016). It is important to note that during a typical explosive event at Etna, only a few kilograms of tephra accumulate on a limited number of streets across the three municipalities. This aspect is closely related to the intensity of the explosive event, the amount of pyroclastic material erupted, and the wind dispersal pattern.

**Table 4. Total tephra mass (kg) computed on main roads selected of Milo, Santa Venerina and Zafferana Etnea for the Etna eruption on 28 February 2021 as derived from the results obtained using Tephra2 and Fall3D models (fixing φ=0.5 and assuming a road width of 6 ± 0.5 m). The total mass on the road network of Milo, Santa Venerina and Zafferana, for three road widths (5.5 m, 6 m, and 6.5 m), is presented in the last three rows. On the right column we include the mass ratio between Fall3D and Tephra2 results.**

| Lava fountains on 28 February 2021 | Total tephra mass ($10^7$ kg) | | |
|---|---|---|---|
| Location | Tephra2 (5.5, 6, 6.5 m-width road) | Fall3D (5.5, 6, 6.5 m-width road) | Mass Ratio |
| Milo-Via V. Bellini | 2.4, 2.6, 2.9 | 1.3, 1.4, 1.6 | 0.5 |
| Milo-Corso Italia | 1.4, 1.6, 1.7 | 1.7, 1.9, 2.1 | 1.2 |
| S. Venerina-Via G. Mazzini | 0.02, 0.2, 0.2 | 0.01, 0.1, 0.1 | 0.5 |
| S. Venerina-Via D. Galimberti | 9.7, 10.6, 11.5 | 9.8, 10.7, 11.6 | 1.0 |
| S. Venerina-Via Stabilimenti | 0.1, 0.4, 0.5 | 0.2, 0.8, 1.0 | 2 |
| Zafferana E.-Via Libertà | 2.7, 3.0, 3.2 | 3.9, 4.3, 4.6 | 1.4 |
| Zafferana E.-Via Zafferana Milo | 1.0, 1.1, 1.2 | 0.3, 0.3, 0.3 | 0.3 |
| Zafferana E.-Via delle Rose | 1.6, 1.8, 1.9 | 0.4, 0.5, 0.5 | 0.3 |
| Total mass on the municipality's road network ($10^7$ kg) | | | |
| Milo | 170.0, 185.4, 209.1 | 131.4, 143.3, 155.3 | 0.8 |
| Santa Venerina | 129.6, 141.4, 153.2 | 144.1, 157.2, 170.3 | 1.1 |
| Zafferana Etnea | 659.1, 719.0, 779.0 | 165.2, 180.2, 195.2 | 0.3 |

## 4.2. Tephra mass on specific road-points

We investigate how the tephra load, derived from both models, can be used to assess the accumulated tephra mass on the road network for the selected municipalities, assuming a cell size of (5×5) metres in the interpolated tephra load map. It is important to highlight that deposited tephra causes disruptions on main roads, specifically in terms of the kilometres of roadways that may face critical driving conditions. The location of the Etna volcano, along with the prevailing westerly and north-westerly winds at high altitude, favours the tephra fallout and dispersal primarily toward east (31%), southeast (35%)

and north-west (29%), and only (6%) directed towards south. These patterns are derived from the analysis of ERA5
reanalysis wind data during 39 eruptive events that occurred in 2021. These results are consistent with the historical
statistical distribution of wind direction and velocity from the 1990-2007 period at altitudes between 5 and 10 km, as
derived from meteorological forecast data (Barsotti et al., 2010; Scollo et al., 2013). The deposited tephra mass, derived
from two models, is computed by selecting eight road-points (shown in Figure 1) across different roads in the municipalities
of Milo, Santa Venerina (Sven), Zafferana Etnea (Zaff) and Giarre municipalities: provincial roads (SP41, SP92, SP8 and
SP59); the highway (E45); and the state road (SS114). To increase the number of road-points on the southeast flank of Etna,
we also include road-points from the E45 and SP114 in the municipality of Giarre. In Figure 5 we show the time cumulative
tephra mass for different median $\phi$ values at specific points in the selected roads, as computed by Tephra2 (Figure 5a) and
Fall3D (Figure 5b). Generally, increasing the median $\phi$ increases the deposited tephra mass and vice versa. Obviously, this
estimate depends on the collection point, because for a given $\phi$, if the mass deposited in the proximal area increases, it
decreases in the distal one, since the total mass deposited is conserved. The highest simulated tephra mass values from both
models are found at SP59 in Milo, with Tephra2 recording values between 2750 and 3375 kg and Fall3D showing results
between 3000 and 4250 kg at the end of the paroxysm sequence. This location is approximately 12 km from the summit of
Etna, making it the closest among the analysed points. In contrast, SP92 in Zafferana Etnea shows lower ash loads, with
Tephra2 reporting tephra load of about 725 to 800 kg and Fall3D indicating results between 250 and 450 kg at the end of the
paroxysm sequence. While Zafferana Etnea is also near Etna (12.2 km away), it is situated further south than Milo. We can
evaluate these results considering the tolerance boundaries, usually considered as more/less five times the estimated values
(Scollo et al., 2007). Indeed, the variability of tephra mass for all the road-points computed with Tephra2 ranges between
limit values of 750-3375 kg, whereas the variability derived from Fall3D is between 250-4250 kg. The time-cumulative
function derived from Tephra2 presents larger steps in concomitance with events of 14 March, 22 and 28 May and 1 July
2021 for SP59 in Milo, E45, SP49 and SS114 in Santa Venerina and SP8 in Zafferana Etnea, whereas E45 and SS114 in
Milo show a larger step during the event on 24 May 2021. During the other events, the tephra mass is not large enough to
generate major steps in the time-cumulative function. For most road-points, the total accumulated tephra mass stabilises at
values between 500 kg and 1250 kg after the paroxysm of 20 July 2021, except for SP59 in Milo. The time-cumulative
function obtained with Fall3D shows larger steps during the events of 28 May, 2 and 27 June, 31 July 2021 for SP59 in
Milo. E45 and SS114 in Milo show a larger step on 27 June 2021, whereas SP41, E45 and SS114 in Santa Venerina exhibit
a large step on 29 August 2021, and SP92 on 17 June 2021. In Zafferana Etnea, SP8 shows a gradual increase in the tephra
mass in the time alternating with long periods of stability. The time-cumulative tephra mass on SP41, E45 and SS114 in
Santa Venerina and SS114 on Milo stabilises between 1325 kg and 2075 kg starting from the beginning of August. Also, in
the case of Fall3D, SP59 in Milo is confirmed as the most impacted road-point due to accumulation of tephra in time where
the time-cumulated tephra increases more than at the other target points, with larger steps, and then stabilises starting from
August. These results suggest that, following the 2021 cluster of Etna lava fountains, the roads of Milo and Santa Venerina
municipalities have been the most impacted from tephra deposition in time.

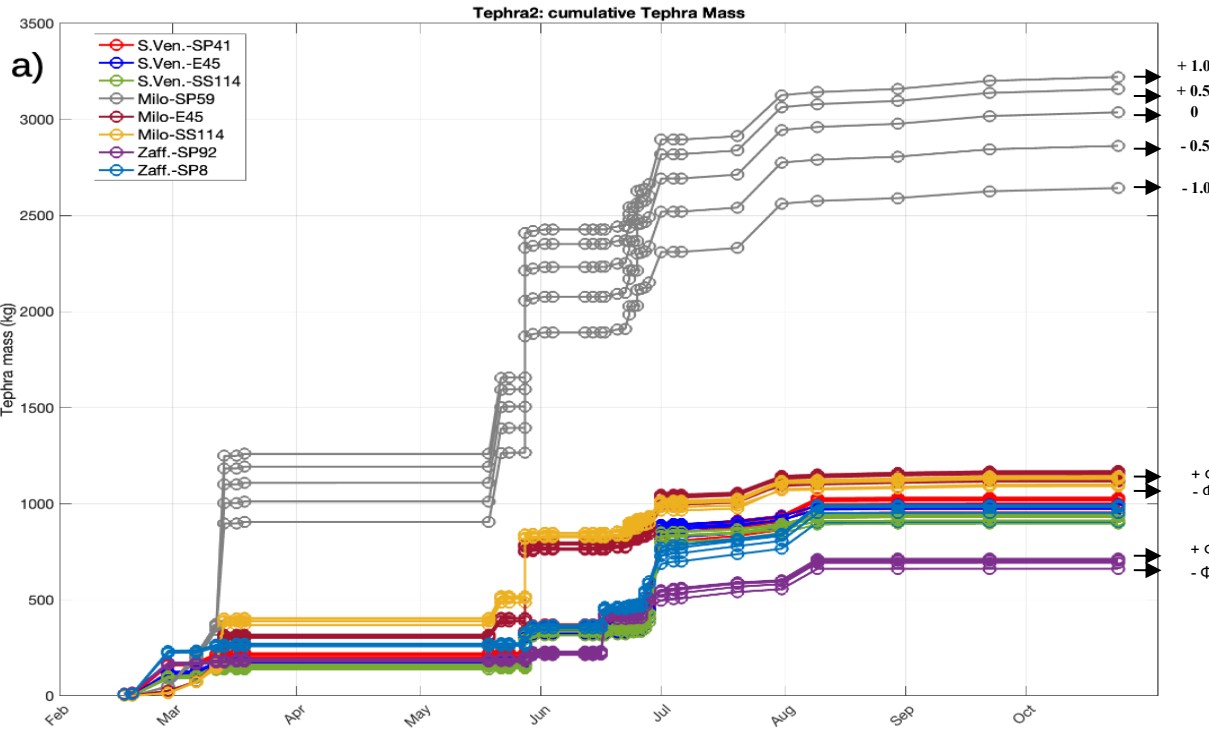


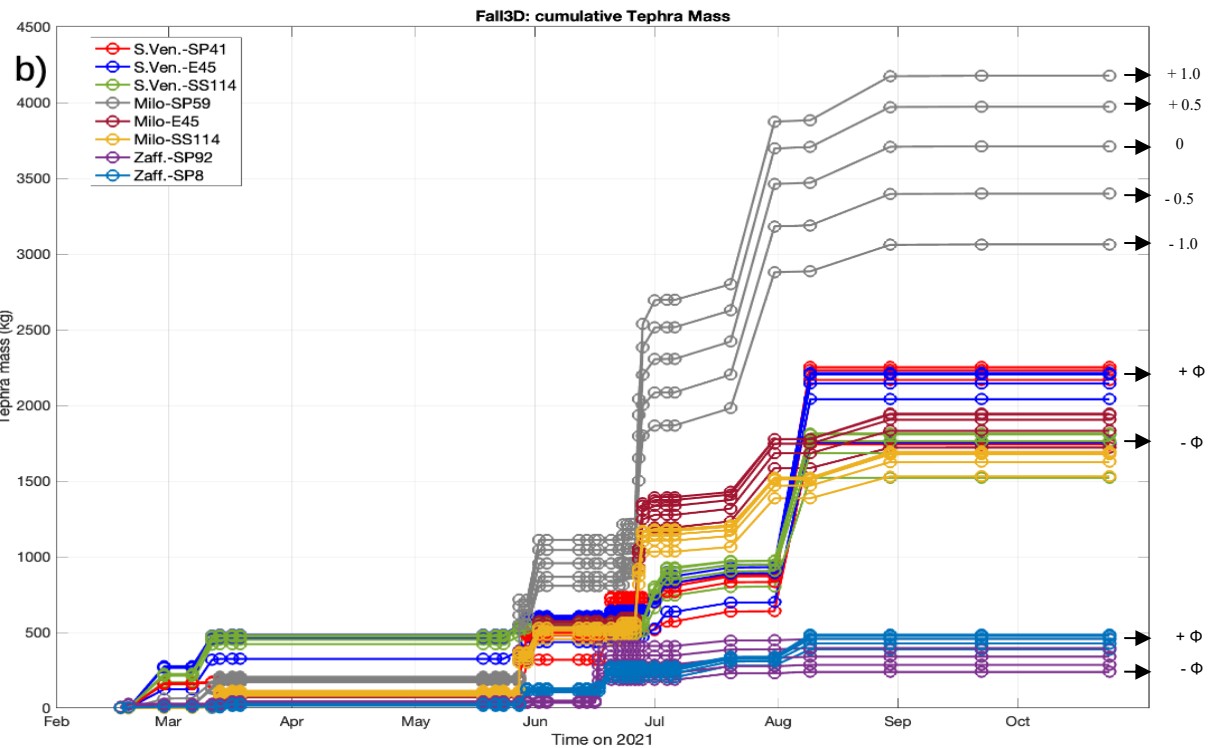


**Figure 5: The time series of cumulative tephra mass simulated by the Tephra2 (a) and Fall3D (b) models for all analysed explosive**
**events at Etna in 2021, which shows plumes dispersing to the east and southeast. Each road-point is represented by a unique**
**colour and symbol. We display the cumulative tephra mass corresponding to each road-point and varying the median value of $\phi$.**
**The larger the mean grain size, the higher the accumulated load for that road-point, as indicated on the right side of each figure.**
**4.3 Total mass accumulated on selected roads**
In this section we quantify the total tephra mass deposited on selected main roads previously listed in Table 3 for each
municipality. Three panels in Figure 6a, 6b and 6c are related, respectively, to the time-cumulative mass on Milo (a), Santa
Venerina (b) and Zafferana Etnea (c) computed on selected roads of known area. Analysing the panels in Figure 6 we
observe some rapid increases in the cumulative trend of tephra mass mainly for the Fall3D simulations (grey dashed line)
with respect to Tephra2 (dark continuous line) simulations. These rapidly increasing trends are found in Milo on 12 March,
28 May, 27 June 2021, in Santa Venerina on 30 May, 17 June 2021 and in Zafferana Etnea. Generally, all the selected roads
show a large step in the estimated cumulative tephra mass on 19 February, regardless of the model used. The largest step of
total mass is found around June and July, as confirmed by the plots shown in Figure 6a.

































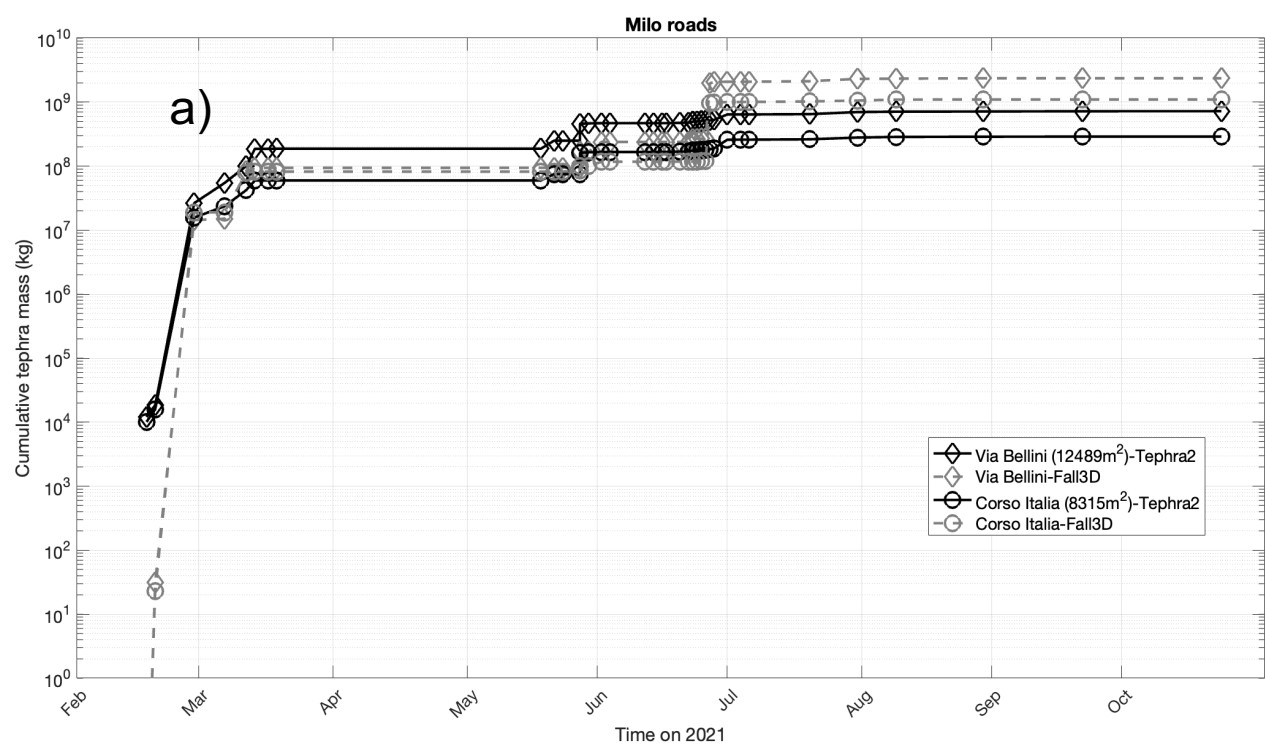

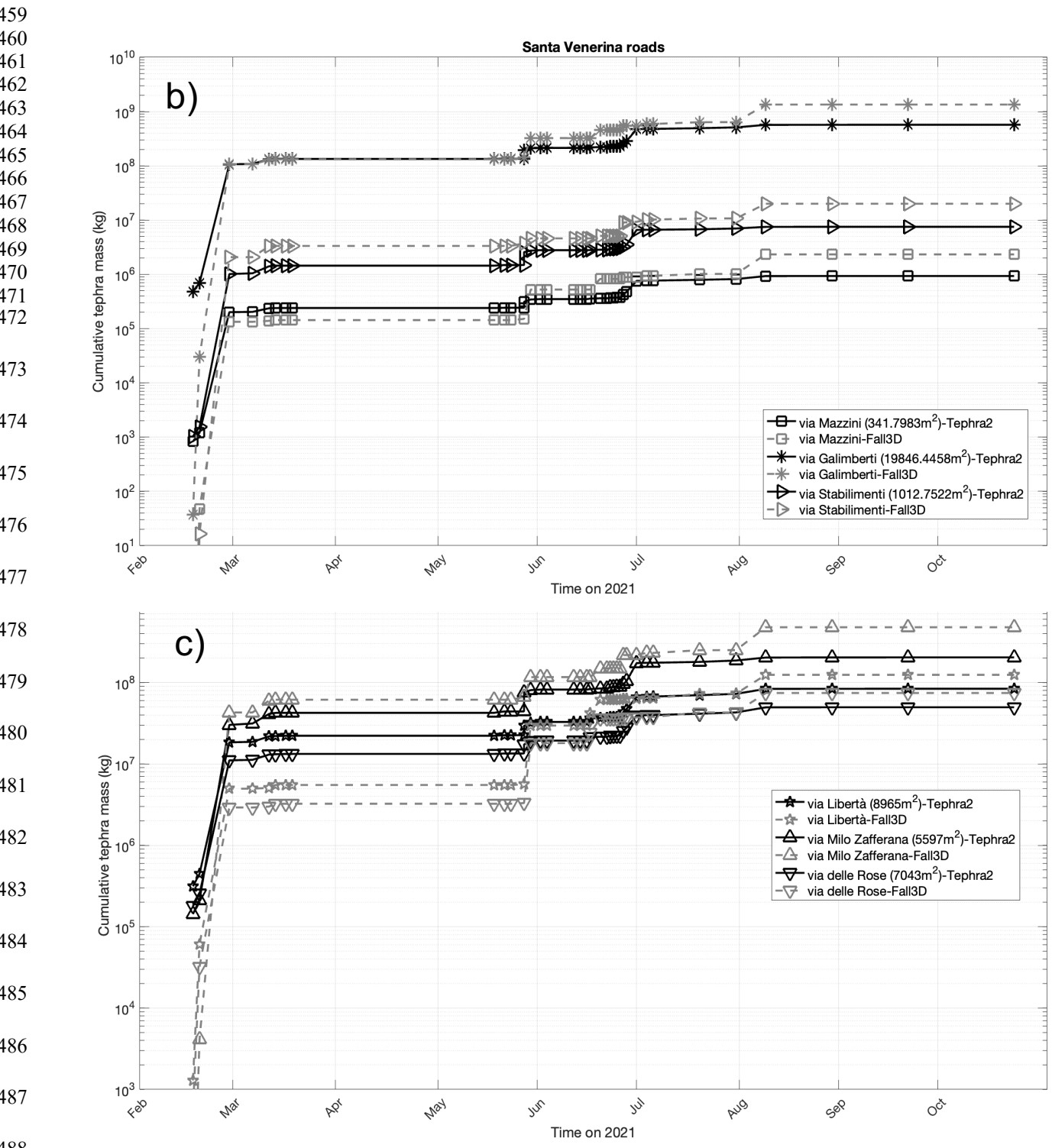

Figure 6: Cumulative tephra mass on selected roads for Milo (a), Santa Venerina (b) and Zafferana Etnea (c), respectively, for all Etna explosive events here analysed and simulated using the Tephra2 (dark continuous line) and Fall3D (grey dashed line)

**models. Each road is plotted with different symbols and identified by the relative area (m², assuming a road width of 6 m), as**
**listed in the legend.**
In Table 5 we summarise the total tephra mass deposited during the 39 Etna events in 2021 for the selected roads in each
municipality. The last three rows show the total tephra mass computed for the complete road network of each town. The
Fall3D estimates are generally larger than those from Tephra2, with the ratio ranging from 1.5 to 3.8 for the roads and from
2.0 to 5.0 for the towns, though the variability can be as much as a factor of five in either direction. We observe that during
about one year of Etna's paroxysms, in the nearest municipalities under examination, the estimated value of total tephra
mass accumulated in the main streets ranges between $10^6$-$10^9$ kg. A difference of at most one order of magnitude in the total
accumulated mass according to the two models is found. It is worth noting that these values are computed under the worst
condition, i.e. without considering the tephra mobilisation due to external factors, such as rain or wind, during the complete
time range, as well as assuming not road cleaning after each event. Therefore, this tephra mass represents a computed
estimate of the total amount of tephra mass that theoretically had to be removed to the roads and disposed of during and
after the 2021 crisis.
**Table 5. Total mass accumulated over 39 Etna events on the selected roads for Milo, Santa Venerina and Zafferana Etnea**
**municipalities, as simulated by Tephra2 and Fall3D models (fixing ϕ=0.5 and assuming a road width of (6 ± 0.5) m). The total**
**mass on the whole road network of Milo, Santa Venerina and Zafferana, for three road widths (5.5 m, 6 m, 6.5 m), is in the last**
**three rows. On the right column the mass ratio between Fall3D and Tephra2 results.**

| Location | Mass ($10^7$ kg) | | Mass Ratio |
|---|---|---|---|
| | **Tephra2** (5.5, 6, 6.5 m-width road) | **Fall3D** (5.5, 6, 6.5 m-width road) | |
| **Milo-Via V. Bellini** | 66.2, 72.2, 78.2 | 216.3, 235.9, 255.6 | 3.3 |
| **Milo-Corso Italia** | 26.6, 29.0, 31.3 | 100.8, 109.9, 119.1 | 3.8 |
| **S. Venerina-Via G. Mazzini** | 0.09, 0.7, 0.8 | 0.2, 1.8, 2.1 | 2.6 |
| **S. Venerina-Via D. Galimberti** | 52.4, 57.0, 61.9 | 123.7, 134.6, 146.2 | 2.4 |
| **S. Venerina-Via Stabilimenti** | 0.8, 3.1, 3.6 | 2.0, 8.2, 9.7 | 2.6 |
| **Zafferana E.-Via Libertà** | 7.7, 8.4, 9.1 | 11.4, 12.4, 13.5 | 1.5 |
| **Zafferana E.-Via Zafferana Milo** | 18.7, 20.4, 22.1 | 43.7, 47.7, 51.7 | 2.4 |
| **Zafferana E.-Via delle Rose** | 4.6, 5.0, 5.4 | 6.8, 7.4, 8.1 | 1.5 |
| **Total mass on the whole road network of each municipality ($10^{10}$ kg)** | | | |
| **Milo** | 7.3, 8.0, 8.6 | 14.9, 16.3, 17.7 | 2.0 |
| **Santa Venerina** | 0.5, 0.8, 1.0 | 1.5, 2.6, 3.1 | 3.3 |
| **Zafferana Etnea** | 5.2, 5.6, 6.1 | 26.6, 29.1, 31.5 | 5.0 |


 **4.4      Total mass accumulated on the full road network**

Similarly to Figure 3, Figure 7 shows the cumulative tephra load (kg/m$^2$) on the geo-referenced road map within the Milo
(light orange area), Santa Venerina (light pink area) and Zafferana Etnea (light green area) municipalities for all 39 Etna
lava fountains, computed using Tephra2 (Figure 7.a)) and Fall3D (Figure 7.b)) models.

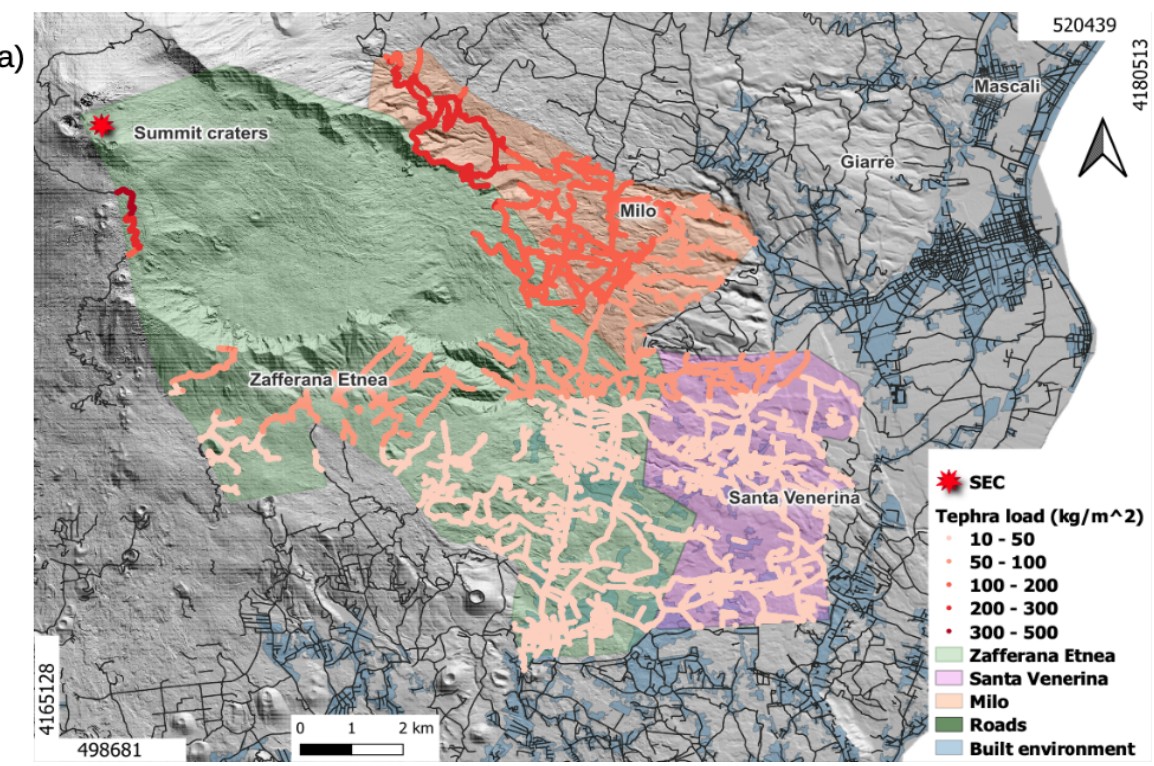


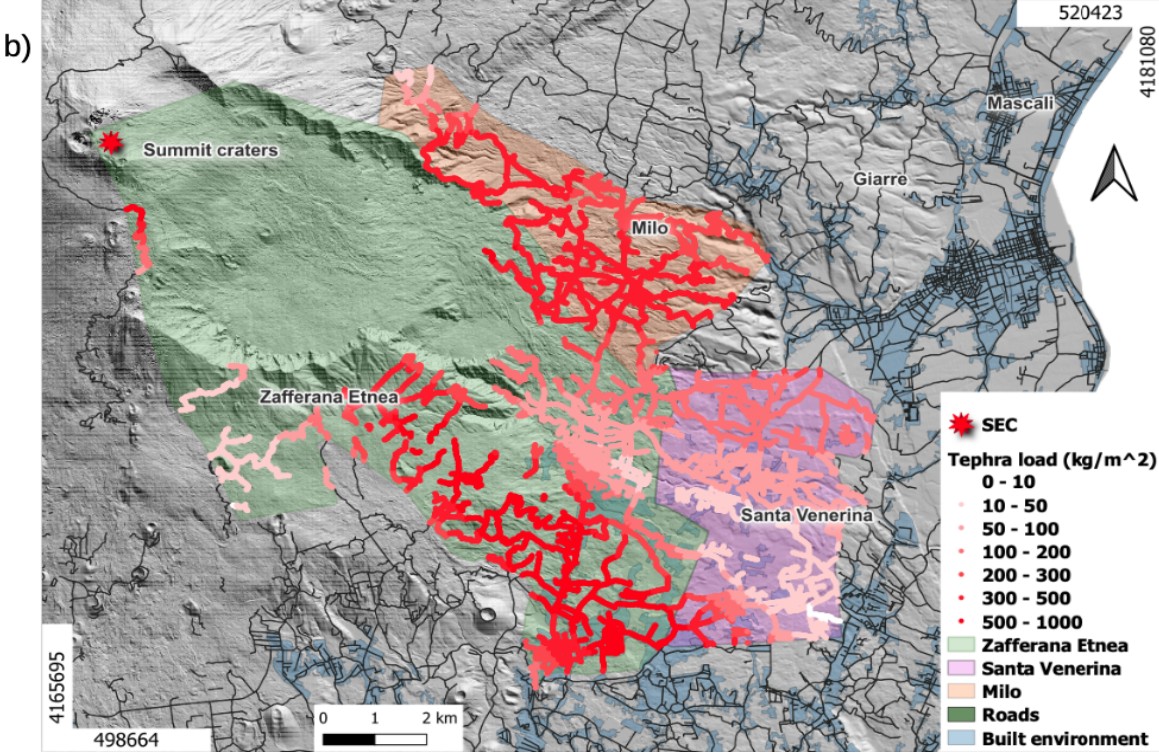



**Figure 7: Cumulative tephra load (kg/m2) for all 39 Etna lava fountains analysed in this work, computed on the whole road network of Milo, Santa Venerina and Zafferana Etnea using (a) Tephra2 and (b) Fall3D models, assuming φ=0.5. The road graph is bold highlighted and in red scale coloured in function of deposited tephra values. Georeferenced maps of road network (dark lines) of the Etna volcano area (shape-files with the road data is publicly available from the Regione Sicilia website: https://www.regione.sicilia.it/).**

Considering the area of each road, we compute the time-cumulative tephra mass (kg) relative to Milo, Santa Venerina and Zafferana, over the whole road network, as computed by both numerical models (Figure 8). Normally, the cumulative tephra mass derived from Tephra2 (dark continuous line) after an initial rapid growth tends to stabilise, in contrast with the trends obtained using Fall3D (grey dashed line), which present rapid increases in estimates throughout the sequence. These leaps in Milo are for the events on 28 May, 23 June (17:40-19:00 UTC) and 27 June 2021, in Santa Venerina on 28 February, 23 and 27 June 2021, and in Zafferana Etnea on 28 February and 17 June 2021. The maximum value of tephra mass deposited on the whole road network in the three municipalities ranges between $10^{10}$ to $10^{12}$ kg.

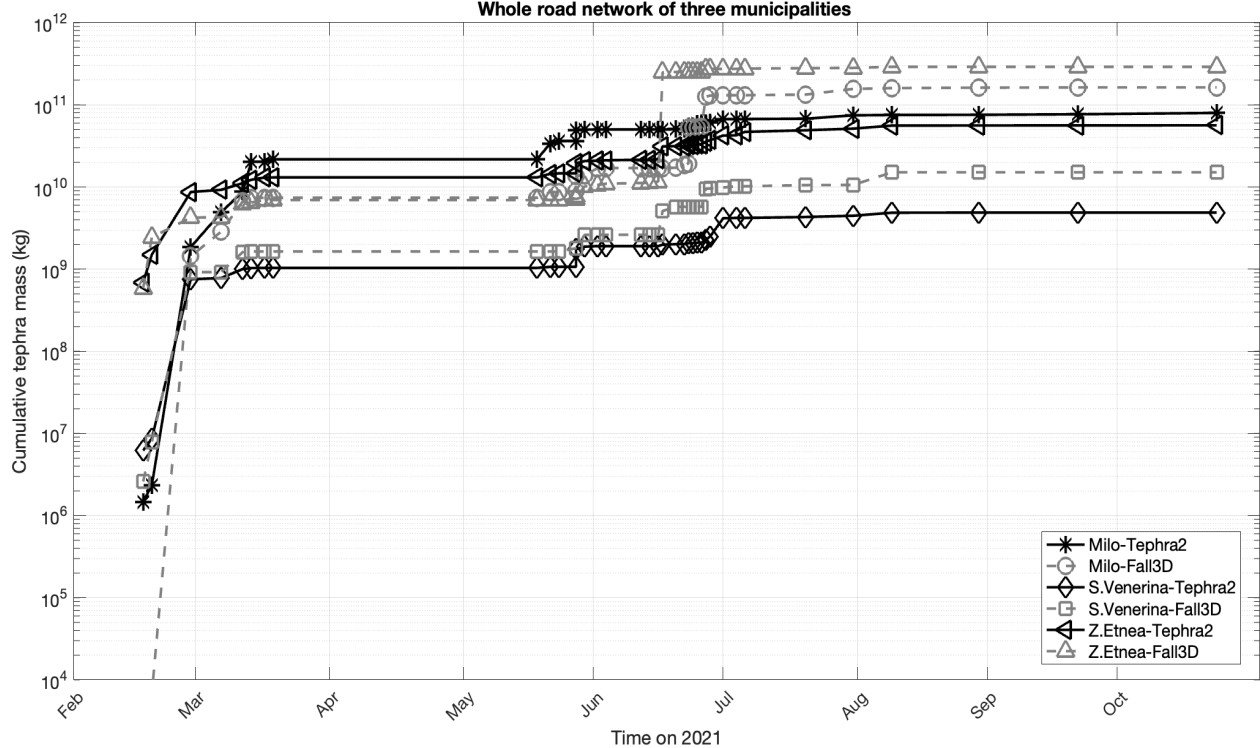

530

**Figure 8: Cumulative tephra mass for all the 2021 Etna lava fountains analysed in this work, simulated using Tephra2 (dark continuous line) and Fall3D (grey dashed line) models, for the whole road network in Milo, Santa Venerina and Zafferana Etnea.**

### 4.5. Analysis of variability in tephra mass results

The estimates of tephra mass, as a function of the uncertainty variation related to ESPs used as input to initialize both numerical models and the specific limitations of the two dispersal models, are already well-documented in the literature (Scollo et al., 2008b). In the present work, we evaluated the potential mass load and total mass accumulated on the road networks of several target towns. Due to the complexity of the Fall3D model, requiring significant computational resources compared to Tephra2, the number of simulations considered for each analysed eruption was limited to the GSD variability. However, the comparison between the results provided by the two models allows a first analysis of the variability in tephra mass estimates. The evaluation of the results for the case study on 28 February 2021 (Table 2), observed by the XWR and analysed by Pardini et al., (2023), allowed us to confirm Tephra2 as a useful numerical model for rapid assessment of tephra dispersal and deposition on road points, and for quickly evaluating the relative tephra fallout hazard. We need highlight that this analysis is only based on one case and considers singles road points it would show that it might not be a representative sample of the model behaviour. These results are consistent with the distribution of field data in relation to XWR measurements and model outcomes shown in Figure 4a, 4b, and 4c. The Kendall's tau correlation coefficient and the MAPE presented in Table 3 further support our findings. We also highlight the high performance of XWR in measuring these quantities, although these observations are not always available in real-time. By shifting focus from individual road points

to selected roads for the same Etna eruption event on 28 February 2021, we observe that the Fall3D estimates are comparable or double with those of Tephra2, with a ratio between their results from 0.3 to 2. When considering the entire sequence of 39 Etna eruptive events, Fall3D estimates are higher than those of Tephra2 for the main roads in the three municipalities and for the entire road network selected in this work. Furthermore, assuming a variability of road width (±0.5 m) in our computations, we estimate a variability in tephra mass of approximately ± 9%. All of this information allows us to outline a more comprehensive framework for estimating the mass of tephra accumulated on roads, which is essential for managing the impact of volcanic eruptions.

## 5. Discussion and conclusions

In this work we assessed, for the first time, the cumulated tephra mass on the road networks in three selected towns on Etna's eastern flank during several paroxysms that occurred in 2021. This accumulated mass is a theoretical estimate of the amount of material that had to be removed from the roads and disposed of during the 2021 volcanic activity. We have focussed on three target municipalities located on the east flank, i.e. Milo, Santa Venerina and Zafferana Etnea. According to the law at the time of the eruption, such material had to be handled and disposed of as special waste (Art. 35 decreto legge 77/2021). At the time of writing this paper a new law allows the use of this material for other purposes (DA n.8/Gab. 22/04/2024, https://www.regione.sicilia.it) as, for example, building construction. Processing measurements derived from visible and thermal cameras of INGV-OE, and, when available, from the analysis of SEVIRI images and the XWR data, we can retrieve the main ESPs, useful inputs to run numerical models. In this way we can simulate and evaluate the cumulated tephra load on roads in time, and processing these results with the QGIS tool, we are able to identify the roads more exposed to tephra deposition. Specific thresholds of tephra load that can damage the main roads system, and the necessary actions to mitigate the tephra effects are defined and known in the literature (e.g. Jenkins et al. 2015; Bonadonna et al. 2021b; Table 8 in Bonadonna et al., 2021a). We consider that the results of this analysis can be a valuable source of information to support the management of volcanic crises and for planning the reinstatement of road networks after a crisis.

It is known that effective and realistic transport management strategies are essential into volcanic contingency planning in sectors where key infrastructure are at risk, such as the road networks. Evaluating tephra mass using different models allowed us to assess epistemic uncertainty and to estimate the sensitivity of each model to the input ESPs and the variability of the median $\phi$. It is worth highlighting that, in this work, we have neglected the uncertainties in the ESP values (such as $Q_M$, TEM and $H_{TP}$), but this analysis is already available in literature (see e.g., Scollo et al., 2008b). However, these values are affected by various sources of uncertainty, including pre- and post-processing of data, as well as instrument sensitivity and accuracy, all of which can significantly impact the model outputs (see e.g., Mereu et al, 2023). This can lead to larger uncertainties in the simulated tephra load in addition to those related to the different model settings and the physical assumptions implemented in each numerical model. Moreover, in this work we are not considering the effect of rain which can remain trapped in the tephra deposit. Furthermore, depending on the rain's intensity, the road traffic safety can worsen (e.g., by making the transport network particularly slippery) or can be improved (e.g., by washing the road surface from the ash deposit).

As a final consideration, we point out the importance of the validation of the results of tephra load simulation obtained with two different numerical models by comparing their output with ground sampling data (in our case available from Pardini et al., 2023) as well as with the XWR retrievals for the Etna explosive activity on February 28, 2021. The tephra deposit estimations, as listed in Tables 2 and 3, highlight the good agreement among the ground sampling, XWR retrievals and the output of numerical models. This observation makes us confident to use the two different models in evaluating not only qualitatively but also quantitatively the tephra deposited during recent Etna paroxysms. In this way it is possible to provide plausible values of ground cumulated tephra mass on roads and identify which routes in the road network of the target towns may be most impacted in next eruptions.

We consider that the analyses and the results proposed in this paper provide interesting inputs for supporting decision making and crisis management. Indeed, such analyses may support planning for clean-up following volcanic eruptions, which is essential for effective volcanic risk management (Hayes et al., 2019). Post-eruption clean-up of tephra deposits on roads is a widespread and costly activity, both in terms of time and resources, and frequently it is an unplanned activity (Hayes et al., 2019). Combining different cleaning methods, such as sweeping, suction, spraying and air blasting, could help speed up tephra fallout removal from high-priority roads used in the management of these events, especially before markings are fully covered (with thickness ranging from 1 to 10 mm), to ensure safety during cleaning operations. Understanding the economic impact on affected areas (Hayes et al., 2015; Magill et al., 2006) also contributes to better risk management. Geospatial analysis methods are well-documented in the literature for estimating the duration of roads cleaning-up (Hayes et al., 2017), although these operations are influenced by complex interactions between physical factors (e.g. erupted volume, column height, grain-size, wind speed and direction, and rainfall) and social factors (e.g. social priorities, prior planning, previous experience, and infrastructure interdependencies) (Hayes et al., 2015). Moreover, the presented results may support decision makers in different ways e.g. for planning and consequently for better management of a future volcanic crisis due to explosive activity of Etna volcano, as well as for getting valuable information about the order of magnitude of the total mass of tephra available for preparing the subsequent disposal and/or reuse.

**Code/Data availability:** Images and analysis can be requested from the author.

**Author contribution:** Conceptualization, A.G., S.S., L.S., L.M.; Data curation, L.M., M.P., M.S.; Formal analysis, L.M., A.G.; Funding acquisition, A.G., S.S.; Investigation, L.M., S.S., L.S., A.G.; Methodology, L.M., A.G.; Project administration, A.G., S.S.; Resources, A.G., M.P., M.S., S.S., L.M., C.B.; Software, L.M.; Supervision, S.S., A.G., L.S., C.B.; Validation, A.G., S.S., L.S., L.M.; Writing-original draft preparation, L.M.; Writing-review & editing, L.M., A.G., S.S., M.P., M.S., C.B.

**Competing interests:** The authors declare that they have no conflict of interest.

**Acknowledgments:** This work was performed in the framework of the INGV Project "Pianeta Dinamico" (D53J19000170001), funded by MUR ("Ministero dell'Università e della Ricerca, Fondo finalizzato al rilancio degli

investimenti delle amministrazioni centrali dello Stato e allo sviluppo del Paese, legge 145/2018"). Fall3D simulations have been performed thanks to the computational resources of the ADA cluster of Istituto Nazionale di Geofisica e Vulcanologia, Sezione di Bologna. Manuel Stocchi was funded by the "PNIR - Programma Nazionale Infrastrutture di Ricerca" with the CIR01_00013 project. C. Bonadonna was funded by SNSF project #200021_188757.

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
