# Peer review of "Estimating the mass of tephra accumulated on roads to best manage"

_EGUsphere, 2024_

## Referee Comment (RC2)

580

[referee-annotated manuscript omitted]

---

## Author Comment (AC1)

We thank the reviewer for the useful comments and suggestions and believe that now this manuscript has been significantly improved. Please, find hereafter our replies to all your comments highlighted in red, whereas all modifications in the main text are in yellow. Thank you for having considered this manuscript.

Dear editor,

in this contribution, Mereu et al. address the problem of estimating the load of tephra accumulated on roads as a consequence of explosive activity at Mt. Etna, Italy. The authors analyze data from 39 explosive events during 2021 in order to obtain a set of inputs to run numerical simulations with the widely used programs Tephra2 and Fall3D. Numerical results are post-processed taking into account the road network in order to quantify the mass of tephra accumulated on critical points around Mt. Etna (from a crisis management perspective). In general, the text is well-written, but some parts are a little bit redundant. On the other hand, a lot of data related to figures (e.g. color scales) are indicated in the main text, and I think they should be restricted to the caption. In addition, some redundant methodological explanations are included in the results section and is not always easy to follow the structure of the text. The state of the art and the addressed problem are introduced in the first section in a clear manner, and this problem is correctly addressed in the manuscript (although a few references are suggested below). The methodology is clear and responses to the introduced problem. Discussion and conclusions are effectively based on results, and presentation of results is accompanied by descriptive figures that effectively display the main results highlighted in the manuscript.

However, I would like to raise the following issues that I think must be addressed before publication:

1. I wonder how efficient is the clean-up of roads due to natural reasons (meteorological phenomena) and effects related to routine human activities (different from cleaning-up activities). The authors describe this point in L348-352 as a limitation of the adopted methodology. This is ok, but it is still critical, in order to understand the validity and significance of results, to have an idea of the order of magnitude of the "natural" clean-up velocity of roads, and thus a larger discussion about this point is needed. For instance, when we analyze the curves presented in Figs. 4 and 5, how are these slopes compared to the expected natural clean-up of roads? Are there differences in the natural clean-up velocity as a function of elevation or vehicles circulation?

>> We have stressed this point highlighting how a combination of different cleaning methods is necessary to speed up the ash removal from high priority routes used in the management of these crises.

2. How comparable are estimates of H_TP derived from XWR, ECV frames and SEVIRI data with respect to those computed from ENT images? Are there examples for which all the methodologies have been applied simultaneously?

>>The H_TP estimate derived from different sensors (XWR, ECV frames, and SEVIRI) shows a time trend that is quite comparable to those described in previous works (e.g., Freret-Lorgeril et al., 2021; Scollo et al., 2019). The H_TP estimations are derived from the analysis of XWR observations, ECV frames, and SEVIRI data. Generally, the XWR estimates show values slightly lower than those derived from ECV and SEVIRI. This difference is due to the sensitivity limitations of XWR in detecting the finest ash particles at higher altitudes, compared to the other two sensors. In contrast, by analyzing the ENT images, whose field of view is mainly focused on the lower section of the volcanic plume (a few kilometers above the volcano vent) and not on the entire cloud, we identify the Incandescent Jet Region, which is considered a proxy for the lava fountain (Mereu et al., 2020).

Hence, this is not a direct measure of the H-TP but only of the mass eruption rate, having consequently higher errors.

3. In general, model limitations should be described better.

>> We have added some texts and references regarding the model limitations. Limitations, in fact, have been already analyzed in literature (Scollo et al. 2008B; Folch et al., 2012, 2016).

4. I identified the following citations issues:

a) In L80, is it Bonadonna et al. 2021a or 2021b?>>done

b) Bonadonna et al. 2023 is not present in the reference list.→Bonadonna et al., 2021a>>corrected

c) Guobadia et al. 2021 is not present in the main text.→Bonadonna et al., 2021a>>corrected

d) Macedonio and Costa 2012 is not present in the main text.>>cancelled

e) Pardini et al. 2023 is not present in the main text.>>this work is cited in par.3.1

All in all, I recommend publication of this manuscript in EGUSphere after minor but essential revisions. In the following lines, I include a set of detailed and editorial comments and suggestions. Please note that my mother-tongue is not English.

Alvaro Aravena

Detailed and editorial comments:

L13: I suggest to delete ", which can require a rapid clean-up".>>done

L14: "reduce" > "evaluate and reduce".>>done

L16: I suggest to delete "top".>> it's the definition of acronym Top Plume Height H_TP

L18: I suggest to delete "this analysis".>>done

L18: "mostly" > "significantly".>>done

L20: "the volcanic ash radar retrieval approach able to retrieve" > "a volcanic ash radar retrieval approach that permits us to compute the".>>done

L21: I suggest to delete "top".>>as before

L21: ", grain size distribution of those events" > "and grain size distribution".>>done

L21: "When the" > "When".>>done

L24: "those" > "the computed".>>done

L26: "allowing" > "allowing us".>>done

L27: I suggest to delete "significant".>>done

L28: I suggest to delete "and disposed".>>done

L29: "quick planning and management" > "planning and quick management".>>done

L29: I suggest to delete "possible".>>done

L31-32: Please rephrase. "Quantification of data" is not a piece of information by itself. I think the phrase "of a specific intensity" is not necessary.>>corrected

L35: "poor visibility conditions" is not a problem by itself. I suggest to delete it and include the reference in the part related to "dangerous road conditions".>>done

L48: ", to allow" > "and to allow">>done

L52: I suggest to delete "try to".>>done

L53: "Etna that were more affected" > "Etna, which were affected">>done

L54-55: "hours and sometimes … times a day" > "hours, separated by periods that can last from few hours to few days">>done

L56: Please delete "generally". The word "most" is already present.>>done

L58: "accumulation" > "tephra accumulation".>>done

L61: "only to 39 events which" > "on 39 events that".>>done

L63: "in" > "to".>>done

L70: "those" > "these".>>done

L71-72: Please define MER and HTP in the first mention.>>done

L73: I suggest moving some of the citations to the previous sentence ("… advection dispersion-models"), where you could include references of other diffusion-advection models that consider the same inputs. For instance:

- Tadini, A., Gouhier, M., Donnadieu, F., de'Michieli Vitturi, M., & Pardini, F. (2022). Particle sedimentation in numerical modelling: a case study from the Puyehue-Cordón Caulle 2011 eruption with the PLUME-MoM/HYSPLIT models. Atmosphere, 13(5), 784.

- Takishita, K., Poulidis, A. P., & Iguchi, M. (2021). Tephra4D: a python-based model for high-resolution tephra transport and deposition simulations—applications at Sakurajima volcano, Japan. Atmosphere, 12(3), 331.

>>We have modified and added the new references

And I suggest to keep the specific references associated with the presentation of the codes Tephra2 and Fall3D in the next sentence ("… such as Tephra2 and Fall3D").>>done

L77: "and to assess uncertainties" > "and assess the associated uncertainties".>>done

L80: 2021a or 2021b?>>corrected

L82: I suggest to delete "theoretically".>>done

L81-90: I think parts of this paragraph are a little bit redundant. >>We simplified this paragraph

>>We have reformulated this paragraph, as highlighted in yellow in the main text

L94: "finally," > ", finally,".>>done

L94: "are in" > "are included in".>>done

L101: ", Fig. 1" > " (Fig. 1)".>>done

L106: "that is" > "calculated as".>>done

L108-109: I suggest to delete "integrating the … we retrieve". Otherwise, the enumeration becomes strange.>>done

L110-112: I suggest to include the following reference (I am sorry for the self-reference):

Aravena, A., Carparelli, G., Cioni, R., Prestifilippo, M., & Scollo, S. (2023). Toward a real-time analysis of column height by visible cameras: an example from mt. Etna, in Italy. Remote Sensing, 15(10), 2595.>>We have added this reference

L116: "altitude derived" > "altitude, which is derived".>>done

L124: "specific" > "the following">>done

L132: "of which three of these under examination and" > "of which three are under examination, and".>>done

L134: "symbol" > "symbols".>>done

L138-139: "is derived … previously described" > "is displayed in Table 1".>>done

L139: I suggest to delete "Usually" if you use the expression "not always" in the same sentence.>>done

L140: "the plume" > "of plume".>>done

L141: "derive" > "collect".>>done

L141-143: I suggest to rephrase this part. >>We rephrased this paragraph

>>We have reformulated this part of text (highlighted in yellow) in the paragraph.

L151: "straight" > "straightly".>>done

L151: "Qm estimates XWR-based" > "XWR-based Qm estimates".>>done

L152: "time" > "and iii) time".>>done

L153: I suggest to delete "iii)" and "iv)".>>done

L156: So are the authors considering a constant wind field at different heights?>>done

L160: I suggest to delete ", as available in literature".>>done

L162: "in the Table" > "in Table".>>done

L168: I think the method used in each case to compute H_TP should be indicated. >> This point is described in points a)-i), b) and c) of par. 2.1.1

L176: "2005; Bonadonna et al., 2006;" > "2005, 2006;">>done

L176: "input" > "inputs".>>done

L177-180: I suggest to enumerate using ';' instead of ',' because you are also including some descriptions (and I would use ',' to separate variables and their descriptions).>>done

L180: "the plume" > "and the plume".>>done

L185: "from buoyant" > "from the buoyant".>>done

L186: "cost in computational time" > "computational cost".>>done

L190: "assuming a" > "assuming".>>done

L190-191: I suggest to delete "In particular … respectively". This should be indicated in the caption. >>done

L192-193: I suggest to delete "Isomass … 5 10^3 kg/m2". This should be indicated in the caption.>>done

L198: "The tephra" > "Tephra".>>done

L202-203: "the geo-referenced … on the ground" > "the geo-referenced data of tephra load on the ground in UTM coordinates … resolution of 500 m".>>done

L218-219: I would end the paragraph after "February 2021". This should be indicated in the caption. >>modified

L230: I am not sure that "validation" is the correct word. I suggest "verification".>>modified

L234: "; Table 2 shows also" > ", as well as the".>>done

L235: "first" > "the first".>>done

L239: "These" > "These discrepancies".>>done

L241: "derived on 14 field data" > "on 14 sites".>>done

L261: "equals" > "=".>>done

L261: "greatest" > "their larger".>>done

L271: I think this parenthesis is not necessary.>>done

L272: "specific" > "discrepancies in the".>>done

L272: "It is worth noting" again is a little bit redundant.>>modified

L272-276: I did not understand this part. Please rephrase.>>modified

L280: "in the" > "is presented in the". >>done

L284-286: Please rephrase (or delete). I think it is not needed to explain results and included in the introduction.>>This sentence has been modified and moved in the introduction

L286: "the Sicily" > "Sicily".>>done

L288: "the east … Etna flanks … at south" > "east (31%), southeast (35%) and nordwest (29%), and only 6% towards south".>>done

L289: It this consistent with wind data in the Etna zone during the last decades?>> We stressed this point in the main text (Scollo et al., 2013; Barsotti et al., 2010). Generally the direction of the wind

disperses the volcanic plume towards the east and south. In this analysis we use the data derived from the ECMWF-Ara5 Reanalysis for more reliable information for Etna events.

L289-293: I think this is a methodological explanation.>>We have modified this sentence which is preparatory to Figure 5.

L295: "Each … symbol" should be in the caption.>>modified

L296: "Obviously" > "Obviously,".>>done

L300: "contrast with" > "contrast to".>>done

L314: "is constant" > "stabilizes".>>done

L317: "to the" > "due to".>>done

L332-333: Please rephrase.>>done

L341: "lava fountains" > "explosive events".>>done

L345: "greater" > "larger".>>done

Table 4: I would combine the cells "39 Etna lava … 2021" and "Location", and call it "Location".>>modified

L358: I suggest to delete "what is shown in".>>done

L365: "assuming a" > "assuming".>>done

L374: "2021 and" > "2021, and".>>done

L382: "eruptions" > "eruption(s)">>done

L383: I suggest to delete "the year".>>done

L387: "; at" > ". At".>>done

L389-390: This is a very interesting point that should be highlighted in the introduction, for instance. >>We have highlighted this point also in the introduction.

L393: I suggest to delete ", focusing on the … municipalities".>>done

L401: "intrinsic" > "different sources of".>>done

L401: "data to" > "data, and due to".>>done

L403: "mainly due both" > "related".>>done

L409: 2021a or 2021b?>>done

L409: Bonadonna et al. 2023 is not present in the reference list.>> corrected: Bonadonna et al., 2021a

L432: Different reference formats are present. Please unify them.>>done

L441-442: This reference does not follow alphabetic order.>>done

L498: Guobadia et al. 2021 is not present in the main text.>> corrected: Bonadonna et al., 2021a

L542-543: Macedonio and Costa 2012 is not present in the main text.>>cancelled

L544-545: Pardini et al. 2023 is not present in the main text. >>This work is in par. 3.1

L579-580: This reference does not follow alphabetic order.>>done

L581-582: This reference does not follow alphabetic order.>>done. We have verified all the references and alphabetically ordered.

---

## Author Comment (AC2)

We thank the reviewer for the useful comments and suggestions and believe that now this manuscript has been significantly improved. Please, find hereafter our replies to all your comments highlighted in red, whereas all modifications in the main text are in yellow. Thank you for having considered this manuscript.

The manuscript presents the results of a modelling study aiming at quantifying tephra deposition from lava fountain events on the Etna volcano road network. Authors use two models (Tephra2 and FALL3D) to model the transport and deposition of the tephra released from the volcano and find that both models give similar results when taking into account the sensitivity tied to volcanic quantities (here a factor of 5).

I find this to be an interesting study looking at an important problem. However, I feel that despite its novelty, at the current stage the study feels a bit too simple. I do believe though that some additional simulations and analysis can lead to a much more impactful publication. To this end I have written down some recommendation and a number of concerns about the methodology employed:

1.      As the authors' note, there is no analysis of the impact of the ESP sensitivity. Considering the relative novelty of the work presented, I would emphatically argue that this is a perfect opportunity to look into the sensitivity of the results to the different combinations of ESP values. I feel that including a robust error analysis (see comment 3) and a quantification of the impact, the study will have a much stronger central point.
>> In paragraph 2 and in the discussion we have added the reference to paper Scollo et al., 2008b, where it is described the sensitivity analysis of ESP on results of Tephra2 and Fall3D. Following your suggestion we have added an error analysis on results derived from both models, from the radar and field data.

2.      I am not sure if Tephra2 is the right model to use here as the study focuses on relatively proximal dispersal of unsteady plumes over very complex topography. From the Scollo et al 2019 study, I understand that this is probably due to the fact that Tephra2 is part of the forecast system employed. Is this correct? I think that objectively, FALL3D is a more appropriate model to use, so I would frame this as cross-examination of the Tehpra2 results (necessitated due to the computational constraints) against a more sophisticated model.
>> The reviewer is right. .Tephra2 is the model for the real-time forecast of ash plume dispersion at INGV-OE surveillance system since 2009 but it also includes topography effects. Differently, the Fall3D model is more complex with respect to Tephra2 but due to the computational constraints is less suited for surveillance issues. In this paper, we don't want to explore what is the best model with respect to another but we wish to estimate the uncertainty due to the different physical assumptions and computational constraints. However, comparisons among the results of the two models and field data of 28 February 2021 were also added to estimate the goodness of both models.

3.      Even though the eruption studied here are described in Scollo et al 2019, I feel that a section discussing their main characteristics is warranted.
>> We added a section to describe the main features of these Etna events.

4.      Despite the presentation of ground observations along the radar-derived values, there is no proper error-based evaluation. This is particularly important for the comparison of the two model results. There's a number of error metrics commonly used (RMSE, MAPE, bias etc) along with correlation coefficients such as Pearson or Kendall tau. Considering the nature of concentrations and depositions, the use of the logarithm error might offer a better tool as it penalises both over and underestimation in the same way. A proper error analysis can help put the conflicting model results into better context.

>> We stressed this point and we added a description of the statistical error analysis on the retrievals of tephra mass such as derived from models and from the XWR, and we have compared them with field data (Pardini et al., 2021); further, we have added a new paragraph 4.5 with some considerations.

5.      I'm not sure if I missed information regarding plumes, but how is tephra introduced in the model? Is there a plume model employed, or is it a standard profile or single point release? I know that the representation of tephra concentration along unsteady volcanic plumes is very much an open issue with no proper answer and considering the fact that forecasts do need to be carried out we must accept the use of simplifications, but I think that important information are missing.

>> We added in the paragraph the information about the plume model considered and different assumptions (Lines 206-209 and 212-217).

6.      Expanding comment 4, in general there's a lot of important model configuration information that seems to be missing from the manuscript.

>> We added other references which refer to detailed information about employed models (Lines 216-217 and 223-225).

7.      The estimation of the road width also needs more information. I have added a relevant comment in the pdf version, but in short, as the results directly scale with the road width chosen further information would help make a more convincing case.

>> We specified in the main text that the roads width have been obtained from both (i) randomly sampling some roads and measuring their width, thus obtaining on average a width of 6 m, and (ii) from the legislation as reported in the text. We also added a width variability of +/- 0.5 m to examine in depth this aspect.

8.      This is a nit-pick, but the interpolation method seems computationally inefficient. Is the intermediate step of interpolating everything at 5m really necessary? I feel that directly interpolating over the road network would be more efficient.

>>The reviewer is right, but in this work we opted to interpolate the tephra load over the whole map, as it was still computationally easily feasible. We added a paragraph with a description of uncertainties and results variation.

9.      Second nit-pick – the word "resolution" is used throughout the manuscript instead of "grid spacing". The two are not the same, as resolution refers to the scale of phenomena models are able to explicitly resolve. In the case of transport modelling, resolution is mostly tied to the grid spacing of the meteorological data.

>>The reviewer is right, we corrected it and replaced it with grid spacing.

Finally, there are some minor points, language errors, typos etc. I've highlighted some in the pdf, but the manuscript merits another careful read-through by the authors.

Overall, I think that this is an interesting and novel study that requires some additional simulations to truly reach its potential. My overall recommendation would be publication after major revisions as discussed above. I hope that the authors will find the comments constructive.

Kind regards and best of luck with the revisions.

>>We thank the reviewer for the several corrections and suggestions pointed out in the pdf, that have greatly improved the manuscript quality. We have changed the text according to each recommendation, and the various corrections are shown in the main text highlighted in yellow.

---

## Author Response (AR2)

We sincerely thank both reviewers for their last valuable comments and suggestions, which have significantly contributed to the improvement of this manuscript. All modifications in the main text are highlighted in green, as suggested by the reviewers. We greatly appreciate your time and consideration of this manuscript.